# Pharmacological Potential of Kaempferol, a Flavonoid in the Management of Pathogenesis via Modulation of Inflammation and Other Biological Activities

**DOI:** 10.3390/molecules29092007

**Published:** 2024-04-26

**Authors:** Faris Alrumaihi, Saleh A. Almatroodi, Hajed Obaid A. Alharbi, Wanian M. Alwanian, Fadiyah A. Alharbi, Ahmad Almatroudi, Arshad Husain Rahmani

**Affiliations:** 1Department of Medical Laboratories, College of Applied Medical Sciences, Qassim University, Buraydah 51452, Saudi Arabia; 2Department of Obstetrics/Gynecology, Maternity and Children’s Hospital, Buraydah 52384, Saudi Arabia

**Keywords:** kaempferol, health-promoting effects, oxidative stress, inflammation, anti-diabetic effect, cancer therapy

## Abstract

Natural products and their bioactive compounds have been used for centuries to prevent and treat numerous diseases. Kaempferol, a flavonoid found in vegetables, fruits, and spices, is recognized for its various beneficial properties, including its antioxidant and anti-inflammatory potential. This molecule has been identified as a potential means of managing different pathogenesis due to its capability to manage various biological activities. Moreover, this compound has a wide range of health-promoting benefits, such as cardioprotective, neuroprotective, hepatoprotective, and anti-diabetic, and has a role in maintaining eye, skin, and respiratory system health. Furthermore, it can also inhibit tumor growth and modulate various cell-signaling pathways. In vivo and in vitro studies have demonstrated that this compound has been shown to increase efficacy when combined with other natural products or drugs. In addition, kaempferol-based nano-formulations are more effective than kaempferol treatment alone. This review aims to provide detailed information about the sources of this compound, its bioavailability, and its role in various pathogenesis. Although there is promising evidence for its ability to manage diseases, it is crucial to conduct further investigations to know its toxicity, safety aspects, and mechanism of action in health management.

## 1. Introduction

Kaempferol (3,4′,5,7-tetrahydroxyflavone) (Figure 1) is a naturally occurring flavonoid in various plant parts. Leafy green vegetables are a great source of kaempferol, and some of the most abundant sources of this flavonoid include spinach, cabbage, and broccoli. Regarding kaempferol concentration, broccoli comes in at 7.2 mg per 100 g, cabbage at 47 mg per 100 g, and spinach at an amazing 55 mg per 100 g. Blueberries have 3.17 mg per 100 g, but onions have a noteworthy 4.5 mg per 100 g [1].

It is worth mentioning that this compound is known for its antioxidant and anti-inflammatory potential, which are indispensable properties that play a substantial role in the management of pathogenesis. A recent study result reported that diquat treatment led to enhanced intracellular ROS production, increased mitochondrial depolarization, as well as apoptosis, which was convoyed by cell cycle arrest at the G1 phase, disrupted intestinal epithelial barrier function, and reduced cell migration. These activities triggered by diquat were upturned by kaempferol. Moreover, this finding revealed that the protective potential of kaempferol was linked with an enhanced mRNA level of genes associated with cell cycle progression and genes implicated in the anti-oxidant system, enhanced Nrf2 (an anti-oxidant transcription factor), and up-regulated abundance of tight junctions [2]. 

Additionally, a combination of lipopolysaccharide and TNF-α activates the inflammatory response of the rat intestinal microvascular endothelial cells (RIMVECs), causing vascular endothelial growth factors (VEGFs) overexpression. Similarly, the permeability of the gut–vascular barrier (GVB), as well as transepithelial electrical resistance (TEER) and the tubular structure of RIMVEC, were meaningfully affected. Kaempferol, with different doses, reduced the GVB permeability and inflammatory factor secretion and down-regulated the hypoxia-inducible factor-1alpha (HIF-1α) and expression of VEGFs, p-Akt. Moreover, kaempferol may inhibit intestinal angiogenesis in the presence of an Akt inhibitor via regulating downstream signaling of the VEGF/Akt pathways and tube formation [3]. Xuerui Yao et al. reported that supplementation with kaempferol enhanced the blastocyst formation rate. Blastocyst formation as well as quality were meaningfully increased in the H_2_O_2_ (200 μM) treatment group following the addition of kaempferol (0.1 μM). This compound prevented the H_2_O_2_-caused compromise of mitochondrial membrane potential as well as reactive oxygen species generation. Additionally, the amount of autophagy as well as DNA damage in the blastocysts was reduced by kaempferol supplementation in the H_2_O_2_-induced oxidative injury group as compared to controls [4]. In addition, kaempferol improves fibrosis by reducing inflammation, oxidative stress, as well as oxidative cellular damage [5]. 

Anti-cancer potential effects of kaempferol in cancer have been proven through the modulation of cell signaling pathways. This compound enhanced autophagy and decreased cell viability, proliferation, as well as migration, and invasion [6,7,8]. Moreover, kaempferol has potential anti-cancer properties through different pathways, such as inhibiting inducing apoptosis, autophagy, G2/M cell cycle arrest, and caspase83-dependent apoptosis [9,10]. 

This compound has an anti-inflammatory role, and Kong and colleagues conducted in vivo studies to investigate the potential anti-inflammatory effects of kaempferol. The animals were given kaempferol, and at the end of the study, it was observed that the cholesterol levels and arteriolar lesions were significantly reduced [11]. Kong and his coworkers discovered that kaempferol has the potential to be an effective anti-atherogenic agent. Researchers reported that kaempferol medication resulted in a noteworthy reduction in the blood levels of several inflammatory indicators, such as TNF-α, leukocytes, cytokines, IL-1β, intracellular adhesion molecule-1 (ICAM-1), and E-selectin [11]. A study on US adults has shown that consuming flavonoids, including kaempferol, is associated with decreased serum CRP levels. This suggests that flavonoids may play a role in reducing the risk of inflammation [12]. This flavonoid capably disturbed the transactivation of STAT3 as well as inhibited more activation of inflammatory cytokines [13]. Kaempferol showed powerful inhibition of COX-1 and 2 enzymes in vitro cell-free assay systems [14], inhibiting the COX-2 expression by curbing Src-kinase activity caused by UVB exposure [15] and its role in other pathological processes [16]. Thus, the present review appears to be focused on providing an in-depth analysis of kaempferol’s sources, synergetic effects, and role in various pathogenesis by modulating biological activities.

## 2. Methodology

This review article covers the pharmacological potential of kaempferol in numerous pathogenesis. Numerous databases were searched to collect information on kaempferol health benefits through Google Scholar, Web of Sciences, PubMed, and Scopus databases up to February 2024. A literature search was conducted using keywords such as bioavailability of kaempferol, oxidative stress, anti-inflammatory, anti-diabetic, cardioprotective, neuroprotective, anti-cancer potential, anti-bacterial, anti-viral, anti-parasitic, hepatoprotective, anti-fungal, anti-thrombosis, anti-obesity, the role of kaempferol in the reproductive system, the role of kaempferol in the respiratory system, the role of kaempferol in oral/dental health, wound healing effects, synergistic effect, and Kaempferol nano-formulation/ disease. In this article, all English studies in the mentioned index were included.

## 3. Bioavailability of Kaempferol

The pharmacokinetics of flavonoids have been widely studied with experiments conducted on rats and humans. Studies have revealed that modifying the molecular structure of flavonoids can significantly affect their absorption and bioactivity. Moreover, glycosylated flavonoids have demonstrated varying levels of bioactivity in both in vitro and in vivo experiments [17]. A study was performed to examine the absorption, excretion, and metabolism of kaempferol in humans. Studies indicate that Endive can provide a modest dose of kaempferol (9 mg), which the body can absorb. After approximately 5.8 h, the mean maximum plasma concentration of kaempferol is 0.1 μM. Whereas a 7.5-fold interindividual variation in maximum plasma concentration was observed between the highest and lowest values, the pharmacokinetic profiles of most individuals were remarkably consistent. Interestingly, about 1.9% of the kaempferol dose was excreted within 24 h. Most of the subjects showed an early absorption peak, which could be due to kaempferol-3-glucoside. This compound accounted for 14% of the kaempferol in endive, making it an important contributor to the overall absorption of kaempferol [18]. The digestion and absorption of kaempferol were studied after consuming 12.5 mg of kaempferol from broccoli for 12 days. The study found that the rate of kaempferol urinary excretion was 0.9% [19]. The study intended to compare the hepatic and small intestinal metabolism of kaempferol and examine its bioavailability plus the gastro-intestinal first-pass effects in rats. The rats received different doses of Kaempferol through either intravenous (IV) administration at 10 and 25 mg/kg or oral administration at 100 and 250 mg/kg. To investigate gastrointestinal first-pass effects, portal blood was collected after oral administration of 100 mg/kg of Kaempferol. The plasma concentration-time profiles revealed high clearance (about 3 L/h/kg) and substantial volumes of distribution (8–12 L/kg) after kaempferol administration at 10 and 25 mg/kg dosages. The plasma concentration-time profiles following oral kaempferol treatment demonstrated a quick absorption with a Tmax of about 1–2 h. The bioavailability (F) was low at around 2% [20].

## 4. Pharmacological Activities of Kaempferol

Kaempferol, a flavonoid found in plants, fruits, and spices has various biological activities associated with health, such as anti-oxidant, anti-inflammation, anti-diabetes, neuroprotective, anticancer, and other biological activities. The pharmacological activities of kaempferol in different pathogenesis are described below.

### 4.1. Antioxidant Potential

Oxidative stress, inflammation, and apoptosis are crucially interlinked processes that hold immense significance in physiological and pathological states [21]. Oxidative stress occurs when an imbalance exists between producing and accumulating reactive oxygen species (ROS) and antioxidants [22]. Various natural compounds showed a role in the inhibition of pathogenesis by inhibiting or reducing oxidative stress [23,24,25]. It has been demonstrated that phenolic compounds possess several antioxidant qualities, such as the ability to stop the production of reactive species, scavenge free radicals and neutralize them, form chelate complexes with pro-oxidizing metals, and remove or repair damage caused by reactive chemicals [26]. Kaempferol plays a role in disease management through its antioxidant potential and helps cells protect themselves from injury caused by free radicals (Figure 2).

Kaempferol has been found to protect human retinal pigment epithelium cells (ARPE-19) against the harmful effects of hydrogen peroxide-induced oxidative cell damage and apoptosis. This has been achieved by activating specific signaling pathways involving Bax/Bcl-2 and caspase-3 molecules, as evidenced by real-time PCR and Western blot results. Furthermore, it has been demonstrated that kaempferol suppresses the H_2_O_2_-induced increase in vascular endothelial growth factor (VEGF) mRNA expression levels in ARPE-19 cells. Additionally, it has been shown to control the activities of superoxide dismutase (SOD) and reactive oxygen species (ROS), which helps maintain the balance of the antioxidant and oxidation systems in ARPE-19 cells exposed to H_2_O_2_ [27]. Earlier research has established that administering kaempferol to diabetic rats significantly improved their plasma glucose, insulin, and lipid peroxidation products. The study also found that the levels of enzymatic and non-enzymatic antioxidants were restored to almost normal levels. These findings advise that kaempferol has effective antioxidant properties, which are evident in its ability to increase antioxidant status and decrease lipid peroxidation markers in diabetic rats. This may protect against the risks of diabetic complications [28]. 

The amount of oxidative stress was assessed by determining the levels of MDA, ROS, SOD, GSH, and GSH-PX in L2 cells. It was reported that kaempferol has no clear role in ROS, MDA, SOD, and GSH-PX GSH in L2 cells under normal conditions. After suffering A/R, the levels of MDA and ROS in L2 cells were meaningfully increased, while the levels of GSH, SOD, and GSH-PX were meaningfully decreased, proposing that the A/R process produced intense oxidative stress in L2 cells. Moreover, this compound pre-treatment considerably decreased MDA and ROS levels, and increased GSH-PX, SOD, and GSH levels, signifying that kaempferol could efficiently decrease oxidative stress induced by A/R. In vivo investigations have shown that kaempferol is useful in decreasing pathological damage, preventing oxidative stress and apoptosis, elevating the expressions of mitochondrial cytochrome c and Bcl-2, and lowering those of cytoplasmic cytochrome c and Bax in the lung tissues of rats following I/R. These results imply that kaempferol may be helpful in shielding lung tissues from the damaging effects of I/R [29]. A study was performed to examine whether kaempferol affects oxidative stress and inflammation in the heart, liver, and lungs after hemorrhagic shock. The SOD activities and MDA levels in the liver, heart, and lung were measured to examine the role of kaempferol on oxidative response following hemorrhagic shock (HS). It was found that MDA levels were meaningfully higher in the HS groups as compared to the Sham group, whereas SOD activities were diminished in the HS groups as compared to the Sham group. The injection of kaempferol following HS exhibited no effects on MDA levels in the heart, liver, and lung, and on SOD activities in the lung heart, and liver compared with the HS group. However, the kaempferol injection 12 h earlier to the induction of HS caused increased SOD activities and reduced MDA levels as compared with the HS group in the lung, heart, and liver. The pretreatment of hemorrhagic shock mice with kaempferol has shown a significant reduction in plasma levels of TNF-α and IL-6, as well as a reversal of MPO, SOD, and MDA in the heart, lung, and liver. Additionally, an increase in the expression of HO-1 was observed in these same organs. These findings suggest that kaempferol could potentially be beneficial in mitigating the adverse effects of hemorrhagic shock in mice [30].

### 4.2. Anti-Inflammatory Potential

The release of reactive species by inflammatory cells during inflammation can lead to oxidative stress. This highlights the close connection between oxidative stress and inflammation [31]. Natural products and their compounds have been discovered to possess protective properties against inflammation [32] and kaempferol’s role as an anti-inflammatory has been confirmed (Figure 2 and Table 1). Dilip Sharma et al., 2019, reported that kaempferol inhibits hyperglycemia-induced activation of RhoA and decreases oxidative stress and fibrosis (extracellular matrix protein and TGF-β1 expression) in RPTEC and NRK-52E cells. TNF-α and IL-1β levels were noticeably increased in high glucose treated cells as compared to normal glucose state. This compound caused a dose-dependent decrease in TNF-α and IL-1β levels in cells [33]. Moreover, at concentrations of 12.5 and 25 μg/mL, kaempferol has been shown to significantly suppress the release of IL-1β, TNF-α, IL-18, and IL-6 while also inhibiting the activation of NF-κB and Akt in lipopolysaccharide plus ATP-induced cardiac fibroblasts. The findings suggest that kaempferol reduces inflammation in cardiac fibroblasts by suppressing the activation of Akt and NF-κB [34]. Another study finding reported that kaempferol reduced IL-32-induced monocyte differentiation to product macrophage-like cells. Furthermore, kaempferol has been shown to inhibit the activation of p38 induced by nuclear factor-κB and IL-32 in a dose-dependent manner in THP-1 cells. Kaempferol has also been found to improve the production of inflammatory mediators such as TSLP, TNF-α, IL-1β, IL-8, and nitric oxide, induced by lipopolysaccharide, in macrophage-like cells differentiated by IL-32 [35]. The study revealed that the oral administration of kaempferol at a 50 mg/kg dose significantly inhibited the antigen-induced passive cutaneous anaphylaxis response in IgE-sensitized mice [36]. Model rabbits fed with a high-cholesterol diet developed noteworthy progression of atherosclerosis. Compared to the control group, the serum levels of blood lipids, IL-1β, TNF-α, and MDA were significantly elevated, while SOD levels were reduced in the model rabbits. The gene and protein expressions of VCAM-1, E-sel, ICAM-1, and MCP-1 in atherosclerotic aortas showed a significant increase in the model group. However, in comparison to the model rabbits, the levels of IL-1β, TNF-α, and MDA decreased significantly, while serum SOD activity increased. Additionally, the gene and protein expressions of VCAM-1, MCP-1, E-sel, and ICAM-1 in the aortas showed a significant decrease with the administration of kaempferol [11]. Kaempferol demonstrates the anti-inflammatory effect through suppressing the translocation of CagA as well as VacA proteins and leading to pro-inflammatory cytokines down-regulation [37].

It has been reported in earlier studies that kaempferol has the capability to hinder the growth of both unstimulated and IL-1β-stimulated RASFs. Additionally, it can also lower the production of MMP-1, COX-2, MMP-3, and PGE2, which are induced by IL-1β. Additionally, it has been reported that kaempferol can inhibit the activation of NF-κB and the phosphorylation of ERK-1/2, p38, and JNK. These proteins are known to play a role in the inflammatory response in RA. Based on these findings, it has been suggested that kaempferol may have the potential as a therapeutic agent for the treatment of RA [38]. It has been reported that at doses ranging from 1 to 20 μmol/L, kaempferol has a dose-dependent effect on the reduction in TNFα-induced expression of epithelial intracellular cell adhesion molecule-1. Additionally, kaempferol has been found to inhibit eosinophil integrin β2 expression, which may hinder the eosinophil–airway epithelium interaction. It has also been reported that kaempferol has the ability to reduce TNFα-induced airway inflammation by decreasing monocyte chemoattractant protein-1 transcription. Moreover, oral administration of kaempferol decreased OVA challenge-elevated expression of eotaxin-1 and eosinophil, major essential proteins, through the blockade of NF-κB transactivation [39]. Kaempferol has shown promising results in increasing the expression of FOXP3, a protein-coding gene expressed in Treg cells, and amplifying mRNA levels of FOXP3 and IL-10 in Treg cells. This suggests that kaempferol may have the potential as a treatment option for autoimmune diseases [40].

**Table 1 molecules-29-02007-t001:** Anti-inflammatory potential of kaempferol with different mechanisms. The downward pointing arrow indicates decrease.

Study Types	Doses	Mechanism	Outcome of the Study	Refs.
In vitro	10, 50 Μm	TNF-α & IL-1β ↓	Kaempferol was found to have a dose-dependent effect on decreasing the levels of TNF-α and IL-1β in these cells	[33]
In vivo	12.5 and 25 μg/mL	TNF-α, IL-1β, IL-6, and IL-18 ↓	It has been reported that treatments with kaempferol have the ability to inhibit the release of inflammatory cytokines	[34]
In vitro	0.02, 0.2, and 2 mg/mL	IL-1b, TNF-a, and IL-8 ↓	In a study on THP-1 cells, it was found that pretreatment with kaempferol significantly inhibited IL-32-induced proinflammatory cytokine production	[35]
In vivo	150 and 30 mg/kg	TNF-α, IL-1β, ICAM-1, VCAM-1 ↓	Gene and protein expression of inflammatory molecules was modulated by kaempferol and it shows an anti-atherosclerotic effect	[11]
In vivo and in vitro	10 or 20 mg/kg 1–20 mmol/L	NF-kB ↓	It was found that OVA challenge increased nuclear NF-kB. However, this activation was disrupted by the administration of kaempferol	[39]
In vitro	40 μM	NF-κB ↓	It has been reported that kaempferol has the ability to induce a dose-dependent inhibition of NF-κB activity	[41]
In vitro	5 to 200 μmol/L	iNOS, COX-2 ↓	It was noticed that at all concentrations, kaempferol reduced the levels of iNOS, COX-2, and CRP protein	[42]

### 4.3. Anti-Diabetic Potential

Diabetes is a chronic metabolic disorder that affects a large population worldwide. The current mode of treatment is effective but it also causes adverse effects. In this regard, natural compounds have proven their role as anti-diabetic through the modulation of various biological activities. It is interesting to note that natural compounds such as kaempferol have been found to have anti-diabetic properties (Table 2). The anti-diabetic potential of kaempferol has been evidenced by the modulation of various biological activities (Figure 3). The anti-diabetic potential of kaempferol was examined, and it was reported that both fasting and non-fasting blood glucose levels of diabetic mice were decreased by kaempferol treatment. Further, STZ-induced diabetic mice exhibited severe glucose intolerance, which was improved by the treatment of kaempferol. Kaempferol therapy gradually increased the activity of hexokinase in the liver and skeletal muscle of diabetic mice, while decreasing the activity of gluconeogenesis and hepatic pyruvate carboxylase [43]. The study assessed kaempferol’s protective role against oxidative stress in streptozotocin (STZ)-induced diabetic rats. It was noticed that diabetic rats presented a slowly raised level of plasma glucose on the initial, 7th, 15th, and 45th days, correspondingly, and a reduced insulin level on the 45th day as compared to normal group rats. Kaempferol oral administration or glibenclamide in diabetic rats exhibited a reduced level of plasma glucose as well as an increased insulin level as compared to diabetic control rats. Moreover, lipid peroxidation levels exhibited a noteworthy increase in diabetic rats, while, upon treatment with kaempferol or glibenclamide, these lipid peroxidative markers displayed a substantial decline towards a normal level. The administration of kaempferol or glibenclamide significantly increased the activities of SOD, CAT, GPx, and GST, which had decreased in the tissues of diabetic rats [28]. It was discovered that giving obese mice oral kaempferol at a dose of 50 mg/kg/day—the human equivalent of 240 mg/day for an average 60 kg human—markedly improved blood control. This activity was related to the reduction in hepatic glucose production in addition to improved whole-body insulin sensitivity, without altering body weight gain, food consumption, or adiposity. The same study found that kaempferol treatment enhanced hexokinase besides Akt activity while decreasing pyruvate carboxylase in addition to glucose-6 phosphatase activity in the liver of obese mice [44]. Kaempferol has been shown to improve insulin sensitivity and blood lipid levels dose-dependently. It has also been found to restore insulin resistance and induce changes in glucose disposal. Moreover, kaempferol inhibited the IkB kinase α (IKKα) and IkB kinase β (IKKβ) and phosphorylation of insulin receptor substrate-1 (IRS-1) [45]. A study assessed kaempferol’s ability to prevent diabetic nephropathy caused by streptozotocin. Renal levels of TNF-α and IL-6, cleaved caspase-3, p38, and Bax were reduced by Kaempferol. This suppressed JNK phosphorylation as well as NF-κB p65 transactivation, along with upregulation of Bcl-2. In both control and STZ-diabetic rats, Kaempferol reduced fasting glucose levels, increased fasting insulin levels and HOMA-β, stimulated SOD and GSH levels, reduced the levels of ROS and MDA, and increased the expression of Nrf2 and HO-1 [46].

### 4.4. Hepatoprotective Effects

Liver diseases, a significant health concern affecting almost 10% of the global population, often lead to cirrhosis and liver cancer [49]. The treatment modules used in liver pathogenesis often cause adverse effects. However, there is hope that natural kaempferol has shown promise in inhibiting liver pathogenesis (Table 3). The hepatoprotective potential of kaempferol has been evidenced by the modulation of various biological activities (Figure 3). Previous findings have demonstrated that natural compounds or their bioactive compounds show a role as hepatoprotective [50,51,52]. It has demonstrated hepatoprotective properties, reducing liver damage in acetaminophen-treated rats. It also decreased the hepatic levels of IL-6, TNF-α, and protein levels of caspase-3, and reduced the increase in circulatory serum levels of γ-GT, ALT, and AST in acetaminophen-treated rats. Interestingly, in both the control and APAP-treated rats, kaempferol suggestively increased the hepatic levels of superoxide and dismutase glutathione, decreased MDA and reactive oxygen species levels, downregulated protein levels of Bax and cleaved Bax, and increased protein levels of Bcl-2 and [53]. The alcoholic liver injury mice revealed clear signs of liver injury, including significantly increased levels of lipid peroxidation, oxidative stress, and excessive CYP2E1 expression as well as activity. The mice given different doses of kaempferol demonstrated enhanced anti-oxidative defense activity and a significant reduction in oxidative stress and lipid peroxidation. The hepatic CYP2E1 expression level and activity were considerably reduced by kaempferol administration, suggesting that kaempferol may down-regulate CYP2E1 [54]. Another study reported that mice with INH/RIF-caused hepatotoxicity displayed meaningfully abnormal serum levels of ALT AST and GSP values, and the administration of kaempferol could decrease these values. In mice, kaempferol reduced the elevation of MDA production and suggestively decreased the depletion of hepatic glutathione. In CCl4-induced abnormalities, oral kaempferol therapy at 5 and 10 mg/kg body weight has been demonstrated to significantly improve liver histology and serum markers. The study found that kaempferol not only improved hepatic histology and serum parameters in those with CCl4-induced anomalies, but it also had a significant impact on reducing pro-inflammatory mediators such as IL-1β and TNF-α, as well as iNOS and COX-2. In addition to its anti-inflammatory properties, the study found that kaempferol also had a positive impact on the oxidative status of the liver. In particular, it increased the glutathione content in rats treated with CCl4 and helped to restore the oxidative status by lowering lipid peroxidation and reactive oxygen species levels [55]. Previous studies have reported the role of kaempferol in hepatoprotection and inhibition of liver pathogenesis [56,57,58,59,60,61].

### 4.5. Gastroprotective Effects

It is interesting to note that natural compounds such as kaempferol have been found to have gastroprotective properties (Table 3). The protective effects and probable mechanisms of kaempferol against acute ethanol-caused lesions to the gastric mucosa were evaluated. Results exhibited that kaempferol significantly reduced the ulcer index, increased the preventive index, preserved gastric mucosal glycoprotein, and protected the mucosa from lesions. The study found that kaempferol had a significant impact on reducing MPO activity and levels of pro-inflammatory cytokines such as IL-1β and TNF-α. Additionally, kaempferol was found to improve NO levels. The study suggests that kaempferol’s gastroprotective activity may be attributed to several factors. Firstly, it was found to preserve gastric mucous glycoproteins levels, which play a key role in protecting the stomach lining. Additionally, kaempferol was found to inhibit neutrophil accumulation and MPO activity, which can contribute to gastric injury [62].

### 4.6. Renoprotective Effects

Kidney-associated pathogenesis affects a large population worldwide and is a significant cause of morbidity and mortality. The current mode of treatment causes adverse effects in various ways. In this regard, natural compounds have proven their role as renoprotective through the modulation of various biological activities. Natural compounds have been shown to have renoprotective properties (Table 3). Kidney diseases have become a significant public health concern worldwide due to their association with severe clinical complications [63,64,65]. It is well-established that oxidative stress and inflammation play substantial roles in the development and progression of chronic kidney disease (CKD) [66]. Renal fibrosis, a hallmark of CKD, is characterized by various factors, such as the accumulation of inflammatory cells, injury to renal tubules, and the development of tubulointerstitial fibrosis [67]. It is crucial to address this issue to ensure better health outcomes for individuals affected by such conditions. Kidney disease is a very concerning health issue that, unfortunately, leads to high morbidity and mortality rates even with the availability of advanced diagnostic and management tools [64,65,68]. It is essential to look into more effective ways to prevent and treat the progression of acute kidney injury and chronic kidney disease [64,65]. There is a need to explore different therapeutic strategies that can help manage these conditions effectively. The renoprotective potential of kaempferol has been evidenced by the modulation of various biological activities (Figure 3). 

The nephroprotective effectiveness of kaempferol and apigenin as dietary supplements in cisplatin-induced renal injury was examined. Findings revealed from MTT assay data, morphology studies, comet, and ROS analysis suggest that CIS 11.36 μM + kaempferol 25 μg/mL and CIS 11.36 μM + apigenin 12.5 μg/mL protect against cisplatin-induced nephrotoxicity. The finding of western blot analysis additionally advises the involvement of NGAL in the apigenin and kaempferol-mediated nephroprotection [69]. Studies have shown that pre-treatment with kaempferol can help reduce the oxidative stress, inflammation, and apoptosis caused by cisplatin. This can lead to a betterment in renal injury and overall kidney functioning. In addition to reducing oxidative stress, inflammation, and apoptosis caused by cisplatin, studies have found that pre-treatment with kaempferol also led to a decrease in cisplatin-induced phosphorylation of p38, JNK, and ERK1/2 in renal tissues [70]. The role of kaempferol in the regulation of blood glucose levels was measured by GLP-1 as well as insulin release during OGTT in C57BL/6 mice. It was noticed that the administration of kaempferol (100 and 200 mg/kg) caused a dose-dependent enhancement in the release of GLP-1. This event was noticeable by a progressive rise in the levels of GLP-1 up to 60 min of treatment, but then again, the highest levels of GLP-1 release were attained 45 min post-administration of kaempferol. Moreover, the insulin levels were meaningfully decreased 19417 by 4.1-fold in diabetic nephropathy (DN) mice when compared to the control group. After 10 days of kaempferol treatment, insignificant changes in GLP-1 as well as insulin levels were noticed. However, after 21 days of treatment, Kaempferol (200 mg/kg) meaningfully increased the GLP-1 as well as insulin levels by 1.5-fold. Kaempferol has shown the potential to improve renal fibrosis and histological changes. It also reduces the expression of DN markers such as CTGF, TGF-β1, fibronectin, collagen IV, RhoA, IL-1β, ROCK2, and p-MYPT1 in DN kidney tissues. An increase in the expression of nephrin and E-cadherin was also found in the same study [47]. 

### 4.7. Neuroprotective Effects

Neurological disease is a significant cause of morbidity and mortality. The current mode of treatment causes adverse effects. In this regard, natural compounds have proven their role as neuroprotective through the modulation of various biological activities such as inflammation and oxidative stress.

Oxidative stress and inflammation can play essential roles in developing various diseases and conditions. Chronic inflammation in the central nervous system has been widely studied and identified as a possible factor in the development and progression of various neurodegenerative diseases [71]. Activated microglia, the immune cells that dwell within the central nervous system, have been recognized as a major factor in the development of neuroinflammation [72]; oxidative stress is a significant factor in aging [73], and it is a universal mechanism causing cell death [74]. Natural compounds such as kaempferol have been shown to have anti-diabetic properties (Table 3). Studies have shown that kaempferol can enhance the activity and levels of antioxidant enzymes like superoxide dismutase (SOD) and glutathione (GSH) [75]. To measure the role of kaempferol, rat pheochromocytoma cells (PC12) and mice were used as neuronal models. In vitro assay-based findings described that kaempferol was revealed to have a protective role against oxidative stress-induced cytotoxicity in PC12 cells. Kaempferol administration significantly upturned amyloid beta peptide (Abeta)-induced impaired performance [76]. ICV streptozotocin (3 mg/kg) administration was performed on the first and the third day of the surgery, and the animals’ memory was examined via passive avoidance tasks. The ICV injections of streptozotocin meaningfully decreased memory retention as well as intact pyramidal cells as compared to the control group. The kaempferol improved the effects of streptozotocin. Results demonstrated that kaempferol can optimize cognitive deficits caused by streptozotocin injections and also has some valuable effects on hippocampal CA1 pyramidal neurons [77]. A study was conducted to evaluate the effects of kaempferol on various factors such as cognitive impairment, lipid peroxidation, hippocampal antioxidants, neuro-inflammation markers, and apoptosis in rat models of sporadic Alzheimer’s disease. The results exhibited a substantial improvement in memory and spatial learning as demonstrated by shortened escape latency as well as searching distance in the Morris water maze in the ovariectomized + streptozotocin + kaempferol group compared with the ovariectomized + streptozotocin group. Kaempferol also showed important increases in brain levels of antioxidant enzymes of glutathione and superoxide dismutase, with a reduction in malondialdehyde and tumor necrosis factor-α. This finding reveals that kaempferol is capable of improving streptozotocin-induced memory impairment in ovariectomized rats, possibly via elevating endogenous hippocampal antioxidants of glutathione superoxide and dismutase and decreasing neuroinflammation. This study advises that kaempferol may be a possible neuroprotective agent against cognitive deficits in Alzheimer’s disease [78]. The research findings indicate that when Alzheimer’s disease flies were exposed to kaempferol, it delayed the loss of their climbing ability and memory. Additionally, kaempferol reduced the activity of acetylcholinesterase and oxidative stress [79].

The study focused on evaluating the neuroprotective effects of kaempferol on hippocampal neuronal cells (HT22) that were exposed to glutamate. The experiment yielded interesting results. It was found that the administration of kaempferol (25 μM) had a significant positive effect on cell viability compared to the control group. Additionally, the neuroprotective properties of kaempferol are believed to be due to its ability to regulate the expression levels of various proteins, including AIF (apoptosis-inducing factor), Bcl-2, Bid, and mitogen-activated protein kinase. Based on these findings, it is suggested that kaempferol could be a promising candidate for pharmacological interventions aimed at both preventing and treating neurodegenerative diseases, such as Alzheimer’s disease [80].

**Table 3 molecules-29-02007-t003:** Role of kaempferol in the management of liver, kidney, and nerve-associated pathogenesis.

**Activity**	**Doses**	**Outcome of the Study**	**Refs.**
Hepatoprotective	250 g/kg, orally	This compound reduced liver damage, inflammatory markers, and liver function enzymes. Increased the antioxidant enzymes	[53]
	10, 20.0 mg/kg	It is worth noting that kaempferol is a powerful compound that can decrease oxidative stress and lipid peroxidation while simultaneously enhancing antioxidative defense activity.	[54]
	5 and 10 mg/kg	Kaempferol hepatic histology and serum parameters and reduced the levels of pro-inflammatory mediators.	[55]
	4.9 mg/kg	Kaempferol reduced TBARS and TNF-*α* levels	[57]
	250 and 500 mg/kg	Kaempferol inhibited the synthesis of collagen and activation of hepatic stellate cells	[58]
	30 and 60 mg/kg	This compound had a positive effect on liver health. Specifically, this compound was found to decrease the levels of alanine aminotransferase and aspartate aminotransferase	[60]
	20 mg/kg/day	Kaempferol may decrease the expression level of LXRα and LPCAT3, thus improving inflammation	[60]
	125 mg/kg	Kaempferol injury through anti-inflammatory, anti-oxidative, as well as anti-apoptotic activities	[61]
Gastroprotective	40, 80, or 160 mg/kg	This compound decreased the ulcer index, completely protected the mucosa from lesions, and decreased pro-inflammatory cytokine activity.	[62]
Nephroprotective	100 and 200 mg/kg	Kaempferol reduced inflammation, oxidative stress, and apoptosis as well as bettered renal injury and its functioning	[70]
	100 and 200 mg/kg	Kaempferol maintains histological changes as well as renal fibrosis	[47]
Neuroprotective	10 mg/kg	Kaempferol treatment increases enhanced acquisition as well as retrieval of spatial memory	[75]
	10 mg/kg	Kaempferol increases antioxidant enzymes whereas it reduces malondialdehyde and tumor necrosis factor-α	[78]
	0, 20, 30, and 40 μM	Kaempferol reduced the acetylcholinesterase and oxidative stress activity and delayed the loss of climbing ability and memory	[79]

### 4.8. Cardioprotective Activity

Cardiovascular disease, such as acute myocardial infarction, has unfortunately become a significant risk factor for human health [81]. Natural compounds have been shown to play a significant role in the management of cardiovascular disease. Many of these compounds are found in numerous foods and supplements. Consumption of these natural compounds in diet can help reduce the risk of cardiovascular disease and improve overall heart health. Kaempferol has been shown to have cardioprotective properties (Table 4). A critical study evaluated the kaempferol on cardiac hypertrophy and the underlying mechanism. Kaempferol has been shown to reduce cardiac hypertrophy caused by aorta banding significantly. This is evidenced by decreased cardiomyocyte areas and interstitial fibrosis, reduced apoptosis, and improved cardiac functions. It has been observed in in vitro experiments that kaempferol has the ability to prevent the activity of the ASK1/JNK1/2/p38 signaling pathway, as well as the enlargement of H9c2 cardiomyocytes [82]. Kaempferol pretreatment (1–10 mg/kg i.p. before DOX administration) showed a dose-dependent recovery of heart and body weights. DOX also caused oxidative stress damage, as indicated by decreased catalase and superoxide dismutase activities in rat hearts and increased lactate dehydrogenase activity in serum. All these actions were dose-dependently bettered by kaempferol, demonstrating its efficiency in counteracting DOX-induced oxidative stress. In addition, in vitro studies have suggested that kaempferol may have mitochondrion-dependent pathways to counteract the cardiotoxicity caused by DOX. It has been observed that this counteraction is achieved through the inhibition of p53 expression and its binding to the promoter region of the Bax proapoptotic gene, while not affecting the Bcl-2 antiapoptotic gene in vitro [83]. Another study reported that kaempferol mitigated hypertrophy and cardiac dysfunction caused by cisplatin. Moreover, pretreatment with kaempferol decreased cisplatin-induced cardiomyocyte apoptosis.

Moreover, cisplatin decreased the expression of BCL-2 and increased the expression of BAX, but such changes initiated by cisplatin were overturned by the treatment of kaempferol. In H9c2 cells, kaempferol intensely reduced cisplatin-induced apoptosis as well as inflammatory response by modulating the STING/NF-κB pathway [84]. Kaempferol pretreatment has been shown to improve the recovery of LVDP and ±dp/dt max, while also increasing the levels of SOD, the GSH/GSSG ratio, and P-GSK-3β. In addition, it has been observed that kaempferol pretreatment led to a decrease in myocardial infarct size, along with decreased levels of cytoplasmic cytochrome C, creatine kinase (CK), cleaved caspase-3, lactate dehydrogenase (LDH), tumor necrosis factor-alpha (TNF-α), and malondialdehyde (MDA) [85].

**Table 4 molecules-29-02007-t004:** The cardioprotective activity of kaempferol.

Types of Study	Model	Dose	Outcome the Study	Refs.
In vivo	Mice subjected to aorta banding	100 mg/kg/d	Kaempferol meaningfully reduced cardiac hypertrophy caused by aorta banding Decreased cardiomyocyte areas as well as interstitial fibrosis, Decreased apoptosis and improved cardiac functions.	[82]
In vitro	H9c2 cardiomyocytes stimulated by phenylephrine	25 μM	Activity of ASK1/JNK1/2/p38 signaling pathway was inhibited by kaempferol This compound inhibited the enlargement of cardiomyocytes.	[82]
In vivo	DOX-induced cardiotoxicity	1–20 mg/kg	Dose-dependent recovery of heart and body weights was noticed by kaempferol pretreatment	[83]
In vitro	DOX-induced cardiotoxicity	0 to μ50 M	DOX-initiated depolarization was also mitigated by kaempferol pretreatment Kaempferol attenuated the DOX-induced DNA fragmentation	[83]
In vivo	Cisplatin-induced cardiac injury	10 mg/kg	Kaempferol mitigated hypertrophy as well as cardiac dysfunction This flavonoid decreased cardiac inflammation	[84]
In vitro	H9c2 cells model	1, 5, or 10 Μm	Kaempferol showed a role in the inhibition of cisplatin-caused cell death Treatment with kaempferol decrease HMGB1, MCP-1, IL-6 and TNF-α	[84]
In vivo	Myocardial Ischemia/Reperfusion Injury	15 mmol/L	Myocardial infarct size and cytochrome C, TNF-α and MDA was decreased by kaempferol	[85]
In vitro	H/R-induced injury	5, 10, 20, or 30 μM	Kaempferol reduces H/R-induced the damages Down-regulation of lactate dehydrogenase (LDH) activity, the increases in anti-apoptotic protein	[86]
In vivo	5-Fluorouracil-Induced Cardiotoxicity	1 mg/kg	Treatment with kaempferol and kaempferol-nanoparticles improved cardiotoxicity	[87]
In vivo	ISO-induced cardiac injury	3 and 10 mg/kg	Kaempferol decrease in serum CK-MB, LDH, troponin-I and lipid profile MDA level and MMP-2 expression and MMP-9 level was reduced by kaempferol	[88]
In vivo	Ischemia-Reperfusion Injury	20 mg/kg	Kaempferol protects against IR injury by reducing apoptosis and inflammation	[89]

### 4.9. Anti-Cancer Potential

Cancer is a multifactorial disease, and unfortunately, it is one of the significant causes of mortality worldwide. It is unfortunate that despite the development of various treatment strategies, cancer remains a substantial cause of death worldwide [90,91]. Cancer treatment modules such as chemotherapy drugs, like doxorubicin, cisplatin, and fluorouracil, and surgical removal are commonly used to treat most cancers. Still, they can also lead to severe side effects and toxicity. Interestingly, many therapeutic drugs used today have their roots in natural resources like alkaloids, taxanes, and flavonoids [92]. There has been a growing interest in using natural compounds in chemotherapy to enhance the effectiveness of anti-neoplastic drugs [93] and inhibit cancer growth through the modulation of cell signaling molecules [94,95,96,97,98]. Kaempferol has confirmed the anti-cancer role by modulating different cell signaling molecules (Table 5 and Figure 4). According to a breast cancer study, kaempferol decreased the expression of IQGAP3 in breast cancer cells. In these cancer cells, kaempferol effectively inhibited proliferation and promoted death while decreasing IQGAP3 expression. It was noticed that upregulating IQGAP3 expression prevented cancer cells from undergoing apoptosis. This was correlated with a rise in the expression of B cell lymphoma 2 (Bcl2) and phosphorylated extracellular signal-regulated kinases 1/2 (p-ERK1/2), as well as a fall in the expression of Bcl-2-associated X protein (Bax). However, this effect was counteracted by the treatment of kaempferol [99]. Furthermore, kaempferol showed a role in the inhibition of MCF-7 breast cancer cell growth, possibly by downregulating Bcl2 expression and inducing apoptosis. Kaempferol is a hopeful cancer preventive as well as therapeutic agent for breast cancer [100]. A recent study based on ovarian cancer reported that kaempferol plays a role in the management of ovarian cancer through the regulation of the cell cycle as kaempferol-induced G2/M cell cycle arrest through the Chk2/p21/Cdc2 pathway and Chk2/Cdc25C/Cdc2 pathway. Further, after the treatment with 40 μM kaempferol, the viability of ovarian cancer cells was reduced to 59% kaempferol-induced apoptosis in cancer cells. The late apoptotic rate of A2780/CP70 cells was meaningfully increased to 23.95% when treated with kaempferol (40 μM) for 48 h [101]. Kaempferol has the ability to inhibit the migration, adhesion, and invasion of human breast carcinoma cells (MDA-MB-231). In addition, it was observed that kaempferol led to a decrease in the activity and expression of MMP-2 and MMP-9 [102]. Kaempferol works synergistically with cisplatin in preventing cell viability of ovarian cancer, and their inhibition on cell viability was brought through inhibiting ABCC6 and cMyc gene transcription. Results reported as cisplatin treatment, with or without kaempferol, did not alter mRNA levels meaningfully for ABCC1, NFκB1, and ABCC5, genes. However, cisplatin reduced cMyc and ABCC6 mRNA levels in a dose-dependent way, with a remaining mRNA level of around 58% at a 80 μM concentration. Kaempferol treatment inhibited cMyc and ABCC6 gene transcription down to 68%. The combination of kaempferol and 40 μM cisplatin prevented ABCC6 as well as cMyc genes mRNA levels down to 65%, and 80 μM cisplatin leads to the lowest mRNA level of 55% for cMyc, though no enhancement is noticed for ABCC6 gene at 80 μM cisplatin concentration. The combination of the two chemicals takes the CDKN1A mRNA level up to 1064.7%. Moreover, the apoptosis assay displayed that the addition of kaempferol (20 μM) to the cisplatin treatment induces the apoptosis of the cancer cells [103]. Moreover, kaempferol time-dependently prevented VEGF secretion and decreased in vitro angiogenesis. Kaempferol decreased ERK phosphorylation and NFκB and cMyc expression. In contrast, it promoted p21 expression. To explore whether ERK signaling participated in kaempferol’s regulation of VEGF, it transfected ovarian cancer cells with ERK1 plasmid and noticed that kaempferol’s inhibition on VEGF transcriptional activation, as reflected by VEGF reporter luminescence, was stopped by forced ERK1 expression in a concentration-dependent way, and a significance was attained in A2780/CP70 cells. This finding proposed that not only does this flavonoid inhibit ERK phosphorylation as well as VEGF expression in ovarian cancer cells, but this prevention of VEGF is also dependent on, at least partially, ERK signaling suppression [104]. It was observed that kaempferol treatment led to time-dependent increases in PTEN expression in bladder cancer cells, especially in those treated with 40 μM kaempferol. Notably, treatment with 40 μM kaempferol resulted in a substantial decrease in the expression of Ser473-phosphorylated Akt. Taken together, these results indicate that kaempferol has the ability to upregulate PTEN expression and prevent pAkt (Ser473) in EJ cells. [105].

### 4.10. Anti-Microbial Effects

The issue of antibiotic resistance, particularly in Gram-negative strains, is quite alarming, especially in hospitals where vulnerable patients are at a higher risk [106]. However, it is not just the hospitals affected by multidrug-resistant (MDR) strains; our food supply also poses a threat due to the rampant use of antibiotics in livestock for infection treatment, growth promotion, and prevention [107]. The extracts from various parts of plants, including roots, stems, fruits, and flowers, are often used to inhibit the growth of microorganisms [108]. It is worth noting that in addition to their antimicrobial activities, it has the potential to be used as preservatives [109,110]. Kaempferol has some promising antimicrobial properties that could potentially be useful in preventing the development and progression of diseases (Table 6 and Figure 5). 

Some of the anti-microbial effects are described here as:I.Anti-bacterial effects

It has been discovered that kaempferol inhibits the action of bacterial efflux pumps and *S. aureus* PriA helicase (SaPriA), which can effectively stop antibiotic-resistant *S. aureus*. This has been shown to increase the effectiveness of antimicrobial treatment [111,112]. Another study result reported that kaempferol showed antibacterial activities against Propionibacterium acnes. Kaempferol 3-*O*-β-(2″-acetyl)-galactopyranoside joined with quercetin showed substantial antibacterial activity through the apoptosis pathway, and it is reported that kaempferol 3-*O*-β-(2″-acetyl)-galactopyranoside was found in clusiacea [113]. The mechanism causing the antimicrobial effect of the antibacterial activity compounds from *H. ascyron* L. was examined. The quercetin and kaempferol 3-*O*-β-(2″-acetyl)-galactopyranoside were noticed from fraction 8 by means of mass spectrometry as well as nuclear magnetic resonance. It was demonstrated that kaempferol 3-*O*-β-(2″-acetyl)-galactopyranoside coupled with quercetin showed noteworthy antibacterial activity through the apoptosis pathway, and it is reported that kaempferol 3-*O*-β-(2″-acetyl)-galactopyranoside was found in clusiacea [114]. The antibacterial activity tests of four flavonoid derivatives, scandenone (1), tiliroside (2), quercetin-3,7-*O*-alpha-L-dirhamnoside (3), and kaempferol-3,7-*O*-alpha-L-dirhamnoside (4), are presented. It has been reported that all the compounds tested exhibit considerable activity against *S. aureus* and *E. faecalis* with MIC values of 0.5 microg/mL. The compounds also show moderate inhibition against *K. pneumoniae* (4 microg/mL), *E. coli* (2 microg/mL), *A. baumannii* (8 microg/mL), and *B. subtilis* (8 microg/mL) [115]. The extract of *B. chinense*, which holds kaempferol-3-*O*-β-d-rutinoside and kaempferol, showed effectiveness against this bacterium [116]. A small molecule called kaempferol has anti-biofilm action and specifically prevents *S. aureus* biofilms from forming. Using fluorescence microscopy and crystal violet (CV) staining, kaempferol (64 μg/mL) prevented biofilm formation by 80%. Kaempferol inhibited the expression of adhesion-related genes, the activity of *S. aureus* sortase (SrtA), and the primary attachment phase of biofilm formation. These results suggest that kaempferol provides a foundation for developing a new anti-biofilm drug that could lower bacterial drug resistance and prevent infections caused by biofilms caused by *S. aureus* [117]. When azithromycin and kaempferol are taken together, as compared to when they are taken alone, ARSA-induced osteomyelitis in rats is decreased by oxidative stress reduction, SAPK and ERK1/2 phosphorylation inhibition, and biofilm formation inhibition [118].

II.Anti-fungal activities

Fungal infections are a cause for concern as they can lead to serious health problems and even life-threatening diseases. They can manifest as acute or chronic illnesses, such as allergic bronchopulmonary aspergillosis or less severe infections like *Candida vaginitis* or oral candidiasis [119]. Dermatophytosis is a fungal infection that touches a significant portion of the population, with an estimated 20–25% of people affected by this condition. It can be a particularly severe type of fungal infection that can cause damage to tissues, organs, and nerves [120]. Numerous natural compounds or bioactive compounds show a role in anti-fungal activity. Kaempferol 3-*O*-b-d-kaempferol 3-*O*-b-d-glucopyranoside and kaempferol-3-*O*-[3-*O*-acetyl-6-*O*-(E)-p-coumaroyl]-b-d-glucopyranoside isolated from *S. hymettia* was noticed to be active against *C. tropicalis*, *C. albicans*, as well as C. *glabrata* [121]. The study focused on determining kaempferol’s minimum inhibitory concentration (MIC) against both the planktonic and biofilm forms of the *Candida* parapsilosis complex. It was observed that the MIC ranges for kaempferol were 32–128 μg ml^−1^. Furthermore, kaempferol demonstrated efficacy in reducing the metabolic activity and biomass of developing *C. parapsilosis* complex biofilms [122]. Kaempferol-(coumaroyl glucosyl)-rhamnoside, present in the Trachyspermum ammi extract, has been shown to potentially inhibit the growth of *Candida* spp. [123]. 

III.Anti-viral activities

Viral infections have been a persistent threat to human health worldwide. Viruses are involved in various types of pathogenesis, including cancer. The vast range of natural products presented delivers an auspicious avenue for the discovery of new and powerful antiviral drugs. Another study based in vitro was performed to examine the antiviral effects of kaempferol against a varicella-zoster virus (VZV) clinical isolate. It was noticed that kaempferol meaningfully inhibited VZV replication without showing cytotoxicity. This compound caused its antiviral potential at the same stage of the VZV life cycle as acyclovir, which prevents VZV DNA replication. Overall, this finding proposes that kaempferol inhibits VZV infection by hindering the DNA replication stage in the viral life cycle [124]. According to the study, kaempferol has been found to inhibit viral penetration and replication stages, resulting in a significant decrease in virus load by 4-fold and 30-fold, respectively. The study also suggests that kaempferol can effectively inhibit virus replication if added within 16 h post-infection (HPI), reducing the number of DNA copies [125]. It is becoming more and more essential to develop effective antiviral treatments. The study found that administering kaempferol at a dose of 15 mg/kg via the intragastric route reduced symptoms associated with H9N2 influenza virus infection in BALB/C mice. In particular, myeloperoxidase activity, pulmonary capillary permeability, lung wet/dry weight, inflammatory cell count, and pulmonary edema were all decreased by kaempferol in the mice.

Additionally, it was noticed that kaempferol increased superoxide dismutase activity while decreasing ROS activity, malondialdehyde formation, and the synthesis of TNF-α, IL-1β, and IL-6 [126]. The study indicated that administering kaempferol at 100 μmol/L completely inhibited the replication of bovine herpesvirus 1 in bovine kidney cells. Furthermore, it was found that kaempferol affects viral replication at the post-entry stages. This suggests that kaempferol has potent antiviral properties, which could be attributed to its inhibition of protein kinase B (Akt) signaling [127]. The study evaluated the effectiveness of kaempferol, kaempferol glycosides, and acylated kaempferol glucoside derivatives in blocking the 3a channel. The results suggested that viral ion channels could be a good target for developing antiviral agents. The study also found that kaempferol glycosides were effective candidates for inhibiting 3a channel proteins of coronaviruses [128].

A recent study described flavonoids as promising natural drugs for the treatment of human herpesviruses (HHVs) infections of the nervous system including alpha-herpesviruses, beta-herpesviruses, and gamma-herpesviruses [129]. Moreover, berberine, an alkaloid, and its role in the inhibition of different pathogenesis, including anti-viral, has been proven. An important study presented the anti-human oncogenic herpesviruses (HOHVs) properties of berberine and mechanisms and pathways induced by this alkaloid through targeting the herpesvirus life cycle as well as the pathogenesis of the linked malignancies [130].

Another study based in vitro reported kaempferol to have vigorous antiviral activity against bovine herpesvirus 1 (BoHV-1) replication. Kaempferol (100 μmol/L concentration) inhibited viral replication in MDBK cells. It chiefly affects the viral replication at the post-entry stages. Moreover, at a concentration of 25 and 50 μmol/L, kaempferol could meaningfully decrease the expression of inflammatory mediators such as macrophage inflammatory protein 1 alpha, tumor necrosis factor-alpha (TNF-α), and interleukin-8 in human promonocytic U937 cell-derived macrophages (dU937) in response to LPS stimulation [127]. In a study, flavonols quercetin 3-*O*-rutinoside and kaempferol 3-*O*-rutinoside derived from Lespedeza bicolor were found to combat HSV-1. They were able to block viral infection by inhibiting viral DNA replication and also exhibited virucidal effects [131]. The antiviral effects of kaempferol against a varicella-zoster virus (VZV) clinical isolate were investigated. It was noticed that kaempferol meaningfully prevented VZV replication without showing cytotoxicity. This flavonoid employed its antiviral effect at a similar stage of the VZV life cycle as acyclovir, which prevents VZV DNA replication. Overall, this finding proposes that this flavonoid inhibits VZV infection by blocking the stage of DNA replication in the viral life cycle [124]. A recent study demonstrates the involved mechanisms by which flavonoids decode their antiviral potentials against herpes simplex virus (HSV). Flavonoids are formidable contenders in the battle against HSV infections as they disrupt key stages of the viral life cycle, such as attachment to host cells, entry, DNA replication, latency, and reactivation [132]. 

IV.Anti-parasitic action

The goal of the study is to determine how kaempferol interacts with hamster neutrophils and impacts *E. histolytica* trophozoites. The hamster model is used to evaluate the susceptibility. The results of the investigation showed that trophozoites’ amoebic vitality was dramatically decreased after 90 min of incubation with 150 μM kaempferol. Kaempferol is an effective drug against *E. histolytica* through the decrease in expression of *E. histolytica*’s antioxidant enzymes. Moreover, studies have shown that kaempferol controls a number of neutrophil processes, such as myeloperoxidase (MPO), reactive oxygen species (ROS), and nitric oxide (NO) [133]. Kaempferol is capable of inducing the appearance of perinuclear and periplasmic spaces that are devoid of cytosolic content, as well as multilamellar structures. Moreover, it has been discovered that kaempferol causes proapoptotic death, which is linked to a partial arrest in the S phase of the cell cycle. On G. duodenalis trophozoites, kaempferol has been observed to have a proapoptotic impact. This entails a partial stop of DNA synthesis without oxidative stress or damage to chromatin or cytoskeletal components [134]. The number of adult and encysted larvae was much lower in the groups that received a combination of drugs. In addition, the thickness of the larvae’s capsular layer decreased and the inflammation of the muscles and intestines significantly improved. When combined with albendazole, kaempferol has demonstrated promise as an anti-trichinellosis drug by reducing inflammation and larval capsule formation [135].

**Table 6 molecules-29-02007-t006:** The anti-microbial potential of kaempferol.

Activity	Doses	Outcome of the Study	Refs.
Antibacterial	0, 30, 60 μg	Kaempferol antibacterial activities against propionibacterium acnes	[113]
Antibiofilm	64 μg/mL	The primary attachment phase of biofilm formation and biofilm formation was inhibited by kaempferol	[116]
	1 μg/mL and 1 mg/mL	The combined treatment of azithromycin and kaempferol showed anti-biofilm activity	[117]
Anti-fungal	0.25 to 256 mg ml^−1^	This compound reduced the metabolic activity besides biomass of growing biofilms of the *C. parapsilosis* complex	[122]
	5, 7, 10, and 15 μg/mL	Kaempferol suppressed VZV replication	[124]
	52.40 Μm	Kaempferol decreases PRV-induced cell death	[125]
Anti-parasitic	150 μM	Kaempferol reduced amoebic viability of trophozoites	[129]
	20 mg/kg	This compound prolonged the survival of infected mice	[136]

### 4.11. Immunomodulatory Effects

Immunomodulation is an interesting process that involves modifying the immune response by administering a drug or compound. Immunomodulators are used to alter immune system components [137]. It is interesting to note that several chemical immunomodulators are being used to treat various disorders but such chemical immunomodulators also cause adverse effects. In this scenario, natural compounds or their bioactive compounds, such as kaempferol, have been observed to inhibit pathogenesis through their immunomodulatory effects (Table 7). In vitro models utilizing human peripheral blood mononuclear cells (PBMC), in addition to human myelomonocytic cell lines derived from THP-1, were utilized to study the effects of kaempferol. It was noted that by inhibiting interferon-dependent immunometabolic pathways, kaempferol may have immunomodulatory effects and immune-suppressive behavior. The data based on the findings suggest that the combined effects of kaempferol on several immunologically relevant targets contribute to its immunomodulatory action [138].

### 4.12. Role in Dental Health 

Oral-associated pathogenesis affects a large number of the population worldwide and is a significant cause of morbidity and mortality. However, the current mode of treatment can cause adverse effects. In this respect, natural compounds or their bioactive ingredients have proven to be effective in inhibiting oral pathogenesis by modulating various biological activities.

The study explored the potential benefits of pretreating dentin with kaempferol to enhance dentin bond stability and nano leakage at the dentin-resin interface. The results suggest that the kaempferol treatment group exhibited a higher bond strength than the control group after thermocycling. Furthermore, the FTIR analysis showed that the kaempferol group had a significantly greater PO4 peak than the control group, indicating a stronger cross-link between dentin and collagen [139]. The potential role of dental health is shown in Figure 6.

The research aimed to examine how kaempferol affects the periodontium and the levels of matrix metalloproteinase and tissue inhibitor of metalloproteinase-2 in gingival tissue. The kaempferol application groups showed some promising results in terms of bone area and attachment loss compared to the control groups. Furthermore, it appears that the kaempferol administration groups had lower gingival tissue MMP-1 and -8 levels than the periodontitis and control groups. These results imply that kaempferol might benefit periodontal health [140]. An investigation was carried out to find out how kaempferol affected the oxidative state of GSH and reactive oxygen species (ROS) in aged gingival tissues. In aged gingival tissues, kaempferol was able to, in a dose-dependent way, lower reactive oxygen species (ROS) levels and raise GSH levels. Furthermore, compared to control samples, kaempferol was demonstrated to dramatically lower the levels of TNF-α and iNOS protein [141].

### 4.13. Role in Eye Health/Disease

The scientific evidence supporting the consumption of plant-based natural products for preventing vision loss and reversing visual impairment is quite substantial [142,143]. These natural products’ anti-inflammatory and antioxidant activities are key factors that contribute to their potential health benefits [144]. The potential role of kaempferol in the maintenance of eye health is shown in Figure 6.

Kaempferol administration improved the amount of keratitis, the recruitment of neutrophils and macrophages, the fungal load in the cornea, and the expression of TLR4 and Dectin-1 in the corneas of mice infected with A. fumigatus. By reducing the corneal fungus load, suppressing the recruitment of inflammatory cells, and downregulating the expression of inflammatory factors, kaempferol improved the prognosis of fungal keratitis in C57BL/6 mice [145] (Table 7). Kaempferol was effective in preventing changes in retina thickness and retinal ganglion cell death in mice experiencing ischemia-reperfusion. It was also found to suppress the activations of the NLRP1/NLRP3 inflammasome, caspase-8, and caspase-3. According to the study’s findings, kaempferol effectively inhibited the NF-κB and JNK pathways, decreasing NLRP1/NLRP3 inflammasomes and caspase-8 to lessen retinal ganglion cell death in acute glaucoma [146]. According to a study finding, human corneal epithelial cells’ (HCECs’) ability to proliferate was significantly inhibited in the model group compared to the normal group, and HCECs’ rate of apoptosis was noticeably higher; however, kaempferol was able to effectively increase HCEC proliferation and decrease HCEC apoptosis. Additionally, a comparison was made between the normal group and the model group’s TNF-α, IL-6, and p38 protein levels and mRNA relative expressions. The outcomes demonstrated that, in comparison to the normal group, these expressions and levels were much higher in the model group [147]. The study result suggests that kaempferol may have protective effects on human retinal pigment epithelium (RPE) against oxidative cell damage and apoptosis induced by hydrogen peroxide. The study also suggests that kaempferol could affect the signaling pathways involving Bax/Bcl-2 and caspase-3 molecules. Furthermore, the study noticed that kaempferol might influence the oxidation and antioxidant imbalanced system in ARPE-19 cells as well as inhibit increased vascular endothelial growth factor mRNA expression levels brought on by hydrogen peroxide [27]. The kaempferol role in high-glucose injury in cells from the retinal ganglion cell (RGC) line was investigated. RGC cells with different concentrations of kaempferol as well as high-glucose. The result designated inhibited lactate dehydrogenase leakage, reactive oxygen species (ROS) level, apoptosis, and caspase-3 activity. Furthermore, cell viability increased in RGC cells that were incubated with different concentrations of kaempferol and glucose compared with glucose alone. Kaempferol (60 μmol/L) raised ERK phosphorylation and vasohibin-1 (VASH1) expression, and inhibition of ERK phosphorylation reversed the influence of kaempferol on the expression of VASH1 in RGC cells with high-glucose injury. Overall, the results propose that kaempferol protected retinal ganglion cells from high-glucose-induced damage through VASH1 and ERK signaling [148].

### 4.14. Anti-Arthritis Effects

Rheumatoid arthritis (RA) is a chronic, systematic autoimmune disease [149]. The global incidence of rheumatoid arthritis is approximately 1% [150]. The development of RA is believed to be influenced by various factors, including genetics, environmental factors, and autoimmune dysfunction [151]. The anti-arthritis potential of kaempferol has been evidenced by the modulation of various biological activities (Figure 6). Kaempferol’s role in managing arthritis has been proven (Table 7). The research discovered that kaempferol inhibits the migration, invasion, and production of matrix metalloproteinase in fibroblast-like synoviocytes (FLSs) associated with rheumatoid arthritis. Moreover, kaempferol prevented the actin cytoskeleton from rearranging during cell migration. Furthermore, kaempferol potently prevented tumor necrosis factor (TNF)-α-induced MAPK activation without changing TNF-α receptor expression. In mice with CIA, kaempferol was found to lessen the severity of arthritis [152]. The goal of the other study was to determine how kaempferol affected the proliferation of RASFs caused by interleukin 1β (IL 1β) and COX, MMP synthesis, as well as prostaglandin E2 (PGE2) produced by synovial fibroblasts in rheumatoid arthritis. Kaempferol has the potential to inhibit the proliferation of rheumatoid arthritis synovial fibroblasts, as well as the expression of certain proteins and mRNA [38]. Kaempferol meaningfully caused decrease in interleukin-1β-stimulated pro-inflammatory mediators in rat osteoarthritis chondrocytes through preventing the NF-κB pathway. These outcomes advocate that kaempferol had substantial anti-inflammatory as well as anti-arthritis effects [153].

### 4.15. Anti-Obesity Effects

Obesity is a significant global public health concern, as it is associated with various health problems. It is generally defined as having a Body Mass Index (BMI) of 30 kg/m^2^ or higher [154]. Research has indicated that dysfunction in adipose tissue may play a significant role in the development of obesity [155]. Phytochemicals are known to have various mechanisms to combat obesity, such as inhibiting digestive enzyme activities, regulating appetite, and reducing the formation of WAT [156]. Additionally, they are also known to increase WAT browning [156]. 

Yifei Bian et al. experimented to explore the benefits of experimental treatment with kaempferol on intestinal inflammation as well as gut microbial balance in an animal model of obesity. In this experiment, a high-fat diet (HFD) was given to C57BL/6J mice for sixteen weeks, during which the kaempferol supplement served as a variable. HFD-induced fat accumulation, obesity, adipose inflammation, glucose intolerance, and metabolic syndrome were the main findings. Moreover, all these metabolic disorders can be improved via the supplementation of kaempferol. In addition, increased intestinal permeability, overexpression of inflammatory cytokines, and infiltration of immunocytes were also found in the HFD-induced mice. This flavonoid supplementation inhibited gut inflammation by reducing the activation of the TLR4/NF-κB pathway, and improved intestinal barrier integrity. Furthermore, kaempferol supplementation was able to counter the dysbiosis related to obesity [157]. A study examined how kaempferol might influence lipolytic and adipogenesis pathways to increase anti-obesity effects. The findings demonstrated that whereas kaempferol (60 Mm) stimulation of pre-adipocytes reduced intracellular lipid accumulation by 39% in mature adipocytes, it inhibited adipogenesis by 62% in pre-adipocytes. According to the study, incubating 3T3-L1 cells with 60 μM, kaempferol decreased Cebpa mRNA expression compared to control cells. Additionally, the gene expression of Pnpla2 and Lipe was upregulated in the same cells treated with 60 μM kaempferol [158]. The objective of this research was to explore the anti-obesity potential of kaempferol in mice that were fed with a high-fat diet. Study findings demonstrated that mice with high-fat diets showed significantly high blood glucose and serum cholesterol levels and body and liver weight gain after eight weeks. On the other hand, kaempferol therapy reduced elevated levels of triglycerides, blood sugar, serum cholesterol, and body weight in addition to liver weight gain [159].

### 4.16. Role in Skin Health

Natural compounds and their bioactive compounds may play a role in preventing skin aging. In this regard, kaempferol plays a significant role in skin health maintenance or preventing skin aging. Unquestionably, the administration of 100 nM kaempferol was observed to considerably ameliorate the cytotoxicity generated by 2-*O*-tetradecanoylphorbol-13-acetate (TPA) on skin fibroblasts and the production of IL-1β. Kaempferol has been observed to inhibit the production of intracellular reactive oxygen species (ROS), which phosphorylate TPA-induced c-Jun N-terminal kinase (JNK). According to the study’s findings, kaempferol could be able to block the signaling cascade that TPA causes in the skin fibroblastic inflammatory response [160]. The study examined kaempferol’s effect on melanogenesis in PIG1 normal human skin melanocytes and its response to oxidative stress. When treated with kaempferol, the mRNA and protein expressions of TYR, the tyrosinase activity, and melanin content of PIG1 cells increased, TRP1, MITF and TRP2 increased, and the phosphorylation level of ERK1/2 increased. Upon the stimulation of H_2_O_2_, kaempferol decreased apoptosis and ROS production and increased the protein expression of HO-1 in PIG1 cells [161]. A study was conducted to investigate the potential impact of kaempferol on the expression of integrins and the stem cell fate of interfollicular epidermal stem cells. Kaempferol positively affected the thickness of the epidermis when added to a skin equivalent. The results of the immunohistological study showed that the expression of integrins α6 and β1, as well as Fdiabethe number of p63- and PCNA-positive cells, were significantly higher in the kaempferol-treated model. Findings have indicated that kaempferol, a type of flavonoid, may have the ability to increase the proliferative potential of basal epidermal cells [162]. The study utilized the Imiquimod (IMQ)-induced psoriatic mouse model to examine whether kaempferol could have potential effects on psoriatic skin lesions and inflammation. According to the study, kaempferol treatment effectively protected mice from developing psoriasis-like skin lesions caused by topical administration of IMQ. Furthermore, the compound was found to decrease the proinflammatory nuclear factor kappa B (NF-κB) signaling in the skin [163] (Table 7).

### 4.17. Role of Respiratory System

Respiratory disease including asthma, bronchitis, COPD, and ARDS affects a large number of populations worldwide and are a significant cause of morbidity and mortality. However, the current mode of treatment can cause adverse effects. In this respect, natural compounds or their bioactive ingredients have proven to be effective in inhibiting respiratory system-associated pathogenesis by modulating various biological activities.

A study on allergic asthma reported that, in the TGF-β1-induced human bronchial epithelial cells (BEAS-2B), the NOX4 expression was reduced with a kaempferol dose increase. The secretions of IL-33 and IL-25 and the NOX4-mediated autophagy were meaningfully decreased by treatment of kaempferol in the TGF-β1-induced BEAS-2B. Furthermore, in the OVA-challenged mice, kaempferol treatment showed a role in the improvement of airway inflammation as well as remodeling via suppressing NOX4-mediated autophagy. This study has found that kaempferol binds NOX4 to perform its functions in the treatment of allergic asthma, which could potentially provide an effective therapeutic strategy for the further treatment of asthma [164]. Human airway epithelial BEAS-2B cells and eosinophils were used in the study to examine how kaempferol affected airway inflammation linked to endotoxins or cytokines. The suppression of LPS-induced eotaxin-1 protein production by kaempferol (1–20 μmol/L) may have been caused by Janus kinase 2 (JAK2) JAK2 signaling. The study also found that kaempferol, at doses ranging from 1 to 20 μmol/L, could dose-dependently reduce TNFα-induced expression of epithelial intracellular cell adhesion molecule-1 as well as eosinophil integrin β2, which hindered the eosinophil–airway epithelium interaction. Moreover, oral administration of kaempferol was observed to reduce OVA challenge-raised expression of eotaxin-1 and eosinophil major basic proteins [39]. A study was performed to explore the ant-ischemia-reperfusion injury (LIRI) mechanism of kaempferol. The study showed that pre-treatment with kaempferol significantly improved the mitochondrial membrane potential, increased cell viability, increased the expressions of Bcl-2 and mitochondrial cytochrome, inhibited the opening of mitochondrial permeability transition pores, reduced the expressions of Bax in addition to cytoplasmic cytochrome c, and reduced the levels of oxidative stress and apoptosis in L2 cells after A/R insult. In rats’ lung tissues after I/R, kaempferol improved pathological damage, raised the expressions of Bcl-2 in addition to mitochondrial cytochrome c, reduced the expressions of Bax besides cytoplasmic cytochrome c, and suppressed the levels of oxidative stress after apoptosis [29]. Kaempferol pretreated mice presented an important reduction in water content in the lungs. This compound pretreatment displayed a decrease in cytokines IL-1β, IL-6, and TNF-α in lung tissue as well as plasma as a comparison to septic mice without pretreatment. Nonenzymatic antioxidant GSH activities and antioxidant enzymes SOD and catalase were increased with the pretreatment of kaempferol in septic mice. Additionally, kaempferol pretreatment reduced the lung tissue nitrite level besides the iNOS level in septic mice. An important decrease in mRNA expression of iNOS and ICAM-1 was noticed with this pretreatment. Mice pretreated with kaempferol followed by sepsis exhibited more arranged alveolar structures and reduced infiltration of cells in histopathological analysis [165]. Oral administration of kaempferol (≤20 mg/kg) obstructed bovine serum albumin (BSA) inhalation-induced epithelial cell excrescence as well as smooth muscle hypertrophy through reducing the induction of COX2 and the formation of PGF2α and PGD2, together with decreasing the anti-α-smooth muscle actin expression in mouse airways. This compound prevented the antigen-induced mast cell activation of cytosolic phospholipase A2 responsive to protein kinase Cμ besides extracellular signal-regulated kinase. Additionally, the antigen-challenged activation of the Syk-phospholipase Cγ pathway was diminished in these flavonoid-supplemented mast cells [166]. 

### 4.18. Radioprotective Effects

Natural compounds have been proven to be effective as radioprotective agents, helping to lessen the damage caused by radiation. A study aims to investigate kaempferol’s potential protective effect against radiation-induced mortality and injury in vivo and in vitro. Kaempferol efficiently enhances the 30-day survival rate after 8.5 Gy lethal total body irradiation. In a study where mice were exposed to 7 Gy of total body irradiation, kaempferol was noticed to offer protection against radiation-induced tissue damage. Kaempferol was observed to inhibit oxidative stress and reduce morphological changes. Overall, it advises kaempferol can protect against gamma-radiation-induced tissue damage as well as mortality [167]. 

The study was performed to evaluate the possible protective effects of kaempferol on submandibular glands (SMGs) of rats exposed to fractionated gamma irradiation. Interestingly, the administration of kaempferol in rats exposed to fractionated gamma irradiation led to the partial preservation of normal gland architecture in submandibular glands. Another study investigated the effects of kaempferol on rats that underwent fractionated whole-body gamma irradiation. The results showed that the rats exposed to radiation displayed degeneration, gland atrophy, vacuolization, and hyperchromatic nuclei in the acini. However, those rats that were treated with kaempferol showed significant preservation of the normal gland architecture, and there was almost no evidence of acinar vacuolation or degeneration [168].

### 4.19. Anti-Thrombosis Effects

Extensive preclinical studies have demonstrated the strong antithrombotic, antiplatelet, and fibrinolytic effects of certain phytochemicals and plant-derived extracts [169]. Flavonoids have been studied for their potential antithrombotic effects [170,171]. The effects of kaempferol on ROS-dependent signaling pathways, NOX activation, and functional responses in collagen-stimulated platelets were examined in a recent study. Interestingly, it was noticed that, in a concentration-dependent manner, kaempferol considerably inhibited the formation of superoxide anion triggered by collagen. According to another study, this compound was found to bind to p47(phox), a key regulatory subunit of NOX, and was able to prevent collagen-induced phosphorylation of p47(phox) and NOX activation. Moreover, kaempferol also showed significant inhibitory effects on platelet aggregation and adhesion in response to collagen. Furthermore, in vivo experiments showed that kaempferol prolonged the thrombotic response in carotid arteries of mice [172]. A pioneer study was performed to examine whether kaempferol affects pro-coagulant proteinase activity, blood clot and thrombin (or collagen/epinephrine)-stimulated platelet activation, fibrin clot formation, thrombosis, as well as coagulation in ICR (Imprinting Control Region) mice and SD (Sprague-Dawley) rats. Kaempferol meaningfully prevented the enzymatic activities of thrombin and FXa by 68 ± 1.6% and 52 ± 2.4%, respectively. Kaempferol also prevented fibrin polymer formation in turbidity. Kaempferol has also been found to protect against thrombosis development in multiple animal models [173] (Table 7).

### 4.20. Anti-Depressant Effects

The occurrence of chronic diseases is a foremost challenge in global public health [174,175]. Depression, a chronic mental disorder, affects around 15 to 20% of people in the world [176]. According to a study, the active ingredients present in medicinal plants have been found to have antidepressant effects. These ingredients are believed to work by neutralizing various stressors, returning monoamine receptor and neurotransmitter levels to normal, and increasing the level of mood-enhancing neurotransmitters in certain parts of the cortex [177,178]. During the experiment, the mice were tested while orally administered KP or quercetin at a dose of 30 mg/kg/day for 14 days. Results of an experiment suggest that flavonoids such as kaempferol and quercetin may have potent anti-depressant effects. The study found that these compounds were able to reduce the immobility time in both the tail suspension test (TST) and the forced swimming test (FST) [179].

### 4.21. Wound Healing Effects

Inflammation and anti-inflammation are crucial in various physiological processes, including hemostasis, removal of harmful microorganisms and damaged tissues, and wound cleaning [180]. If the inflammation phase lasts longer than necessary, it can become a problem and hinder the wound-healing process [181,182]. A critical study was performed to examine the wound-healing effects of kaempferol. According to macroscopic examination, the diabetic excisional and nondiabetic incisional wounds treated with 1% (*w*/*w*) kaempferol ointment for 14 days both showed good wound healing effects (92.12% and 94.17%, respectively). According to a study, wounds treated with 1% kaempferol ointment showed significantly higher hydroxyproline levels than control groups. Specifically, excisional and incisional wounds treated with the ointment exhibited hydroxyproline levels of 2.84 and 2.07 μg/mg, respectively. This flavonoid has shown promising results in promoting wound healing in diabetic and nondiabetic patients [183] (Table 7).

### 4.22. Role in Bone Disease

Osteoporosis is a multifactorial systemic bone disease that is categorized by the destruction of bone microarchitecture, decreased bone mass, as well as increased bone fragility [184]. Natural products and their bioactive ingredients have confirmed their role in the treatment of osteoporosis. The Kaempferol-treated group showed suggestively higher bone mineral density in the trabecular regions (proximal tibia, femur neck, and vertebrae) and lower serum ALP as compared with the ovariectomized rats. The compressive energy of the vertebrae was meaningfully higher in the ovariectomized +Kaempferol treated group compared with the ovariectomized group. Moreover, this flavonoid treatment of ovariectomized rats caused an increase in osteoprogenitor cells and prevention of adipocyte differentiation from bone marrow cells compared with the ovariectomized group [185]. Another study was performed to examine whether kaempferol has a beneficial role in glucocorticoid (GC)-induced bone loss. It was reported that glucocorticoid (GC) was associated with reduced bone mineral density as well as impaired bone microarchitecture parameters. Consumption of this flavonoid induced bone-sparing role in GC-induced osteopenic conditions. Moreover, better callus formation at the site of drill injury in femur diaphysis was noticed with the consumption of kaempferol in animals on GC. Consistent with the in vivo finding, kaempferol caused more expression of osteogenic markers in vitro as well as antagonized the apoptotic effect of dexamethasone on calvarial osteoblasts [186]. Jun Zhu et al. reported that kaempferol increased LPS-induced levels of chondrogenic markers and reduced the level of matrix-degrading enzymes, suggesting the osteogenesis of bone marrow-derived mesenchymal stem cells (BMSCs) under kaempferol treatment. Then again, this compound enhanced LPS-induced reduced expression of lipid catabolism-related genes and suppressed the expression of lipid anabolism-related genes. The Oil red O staining further confirmed the inhibition effect of kaempferol on BMSCs adipogenesis. Furthermore, kaempferol alleviated inflammation by reducing the level of pro-inflammatory cytokines by inhibiting the nucleus translocation of nuclear transcription factor (NF)-κB p65 [187].

**Table 7 molecules-29-02007-t007:** Kaempferol’s role in the prevention of different pathogenesis.

Pathogenesis	Dose	Outcome of the Study	Refs.
Gum disease	10 mg/kg	Greater bone area, less alveolar bone, and attachment loss were seen in the kaempferol treatment	[140]
Glaucoma	100 mg/kg	Kaempferol reduced RGC death and decreased the change in retinal thickness brought on by IOP This drug suppressed the activations of caspase-3, caspase-8, and NLRP1/NLRP3 inflammasome activation	[146]
Retinal Degeneration	50 Μl	Kaempferol prevents retinal cell death and guards against pathological alterations in the retinal tissue	[27]
Arthritis	50, 100, 200 mg/kg	Disease severity in addition to joint swelling was diminished by kaempferol. Moreover, kaempferol attenuated joint inflammation by reducing inflammatory cell infiltration	[152]
	0, 50, 100, 200 μM	Kaempferol inhibits the IL-1β-induced proliferation	[38]
Obesity	0.1%	Kaempferol supplementation decreased body weight and weight gain and reduced weight of epididymal fat	[157]
	200 mg/kg	In mice fed a high-fat diet, kaempferol can prevent obesity and insulin resistance	[159]
Psoriasis	50 & 100 mg/kg	Kaempferol alleviates histopathology and morphological skin lesions Kaempferol treatment groups displayed notably reduced parakeratosis or epidermal thickness and smoother epidermis	[163]
Asthma	20 mg/kg	Airway inflammation and remodeling was improved by kaempferol treatment	[164]
	10 & 20 mg/kg	The administration of kaempferol prevented the increase in eosinophil numbers and recovery to eosinophil counts.	[164]
Radiation-induced pathogenesis	5 &15 mg/kg	This compound showed a role in the protection against radiation-induced tissue damage and mortality	[167]
	10 mg/kg	Cotreatment with kaempferol partially maintains the normal gland architecture	[168]
Thrombovascular diseases	10, 30 μM	Kaempferol appears to have a suppressive effect on the generation of O2• in collagen-stimulated platelets Kaempferol has the ability to attenuate the activation of NOX that is induced by collagen	[172]
Thrombosis	20 mg/kg	The treatment with kaempferol resulted in a 60% survival rate in animals that were subjected to a thrombotic challenge Kaempferol showed an anti-thrombotic effect	[173]
Depression	30 mg/kg	Kaempferol showed a potent antidepressant effect	[179]
Wound	0.5% and 1% weight/weight	This compound showed an effective topical wound-healing agent	[183]
Bone health	5 mg/kg	Kaempferol decreases GC-induced bone loss and increases bone regeneration at fractured sites, therefore emphasizing the positive effect of flavonoids on bone health.	[186]

## 5. Nanoformulation of Kaempferol and Its Role in Inhibition of Pathogenesis

Natural compounds, including flavonoids, have great potential in treating disease but have limitations such as low solubility and rapid removal. Different nano-formulations with better efficacies are being used to overcome such problems. Nanoparticles have become an increasingly popular tool in biomedicine due to their unique chemical and physical properties, which stem from their nanometer size [187,188,189,190]. Various types of kaempferol-based nanoformulations have confirmed better efficacies in the management of different types of pathogenesis (Table 8 and Figure 7). The flavonoids, specifically kaempferol, and corticosteroids, specifically hydrocortisone, were used in the preparation of silver conjugated kaempferol and hydrocortisone nanoparticles (KH-AgNPs) and findings have demonstrated the better antibacterial efficiency of the nanoparticles [191]. Antibacterial analysis against Escherichia coli (ATCC 8099) and Staphylococcus aureus (ATCC 6538) strains showed that Kae-AgNPs displayed superior antibacterial effects than AgNPs or Kae alone. Kae-AgNPs with a low concentration of 2 μg/mL can effectively tackle *E. coli* bacteria [192]. Another recent study designates chitosan/silver nanocomposite synthesis using kaempferol for bactericidal and anticancer activity. Results revealed that Kf-CS/Ag nanocomposite showed potential inhibitory properties against triple-negative breast cancer. Furthermore, synthesized Kf-CS/Ag nanocomposite displayed important and dose-depended antibacterial properties against *P. aeruginosa* and *S. aureus* [193]. The study’s objective seems to be focused on understanding how silver nanoparticles-kaempferol (AgNPs-K) inhibit treated MRSA. AgNPs-K holds antibacterial activity against MRSA, and its mechanism of action is reflected in the gene expression of the biofilm pathway, as well as virulent and glycolysis activity [194]. According to the results of another investigation, kaempferol-coated AgNPs have cytotoxic effects and reduce HepG2 cell viability in a concentration-dependent manner. The percentage of LDH leakage was significantly increased in cells treated with kaempferol-coated AgNPs, indicating a cytotoxic action [195]. The study used the CyQuant assay to measure the anti-proliferation activity of k-AuNPs and found that it induced apoptosis of MCF-7 cells. This was reflected in the increase in the sub-G1 (hypodiploid) population. Additionally, the finding revealed that k-AuNPs inhibit the angiogenesis induced by vascular endothelial growth factor (VEGF), as demonstrated by the chorioallantoic membrane assay [196]. Other studies reported that the synthesized K-AuNCs were effective in targeting as well as damaging the nuclei of cancer cells while being less toxic to normal human cells. These nanoclusters were found to have higher toxicity to a certain type of lung cancer cell (A549), suggesting potential applications for anticancer drug delivery and bioimaging [197]. PEGylated AuNPs-DOX@Kaempferol is a nanomaterial that was created and developed in a groundbreaking study to deliver against colon cancer. The findings showed that when compared to either drug alone, the combination of DOX and kaempferol is more effective at producing a cytotoxic impact. PEGylated AuNPs-DOX@Kaempferol demonstrated a noteworthy decrease in tumor volume in vivo, without eliciting any major adverse effects [198]. Monodispersed gold nanoparticles (KAunp) were synthesized in a rapid reduction reaction catalyzed in kaempferol. Studies showed the KAunps to be very successful against drug-resistant and wild-type pathogens [199]. Moreover, sorafenib and kaempferol encapsulated in PEGylated gold nanomaterial (PEG-AuNPs@SFB/KMF) were developed for a drug delivery system targeting breast cancer. PEG-AuNPs@SFB/KMF were more efficient than sorafenib alone in MDA-MB-231 and MCF-7 cells, and this formulation increased the apoptosis ratio [200]. To create kaempferol nanoparticles (Nps), hydroxypropyl methylcellulose acetate succinate (HPMC-AS) and Kollicoat MAE 30 DP polymers were used as encapsulants. The study found that when KFP-Np (50 mg/kg body weight, 6 weeks) was given orally as a pre-treatment, the elevated serum levels of alanine transaminase (ALT), total bilirubin (TBiL), and aspartate transaminase (AST) were decreased. Furthermore, there was a reduction in the amount of lipid peroxidation (MDA) and a restoration of the levels of antioxidant defense system markers such as glutathione S-transferase (GST), glutathione (GSH), superoxide dismutase (SOD), and catalase (CAT) [201].

The gelatin nanoparticles (GNP) with kaempferol encapsulation (GNP-KA) were synthesized for corneal neovascularization treatment. It was reported that GNP-KA showed the capacity to inhibit the cell viability and function of HUVECs. Also, it was reported that mice’s eyes with corneal neovascularization treated with eye drops comprising GNP-KA once daily for 7 days showed better therapeutic effects with less vessel in-growths in the cornea, compared to the kaempferol solution group by reducing the production of MMP as well as VEGF in the cornea [202]. The study found that nano-formulated water-soluble combretastatin and kaempferol can suppress angiogenesis by inhibiting endothelial cell activation and angiogenesis-suppressive factors. It was also discovered that the combination of combretastatin and water-soluble kaempferol, which had been nanoformulated, performed significantly better than either treatment alone [203]. Kaempferol (KAE) loaded in nanostructured lipid carriers (NLCs) was made to evaluate its role against breast cancer cells. Kaempferol-loaded nanostructured lipid carriers demonstrated moderated cell proliferation. Co-administration of KAE-loaded nanoparticles and paclitaxel into cancer cells meaningfully strengthens the percentage of apoptosis [204].

## 6. Synergistic Effect of Kaempferol with Other Natural Compounds/Drugs

Synergism can be defined as when two bioactive compounds result in similar potential within the body that can produce further effects when used concurrently [206]. In this regard, a study reported that synergistic effects of phytochemicals found that extracts of separate bioactive compounds were not efficient in inhibiting oral cancer, but then by consumption of a multitude of synergistic compounds found within whole foods, a noteworthy therapeutic effect was observed [207]. Previous studies have reported the synergistic effect of kaempferol with other natural compounds/drugs (Table 9 and Figure 8). An experiment was performed using checkerboard assays to explore synergistic actions. It was found that after adding kaempferol, the MICs of colistin were significantly decreased, according to checkerboard tests. The results were interesting; 83% and 17% of the experiment strains showed synergistic and additive effects, respectively. It was observed that the combination of colistin (col) and kaempferol had a powerful synergistic effect on most of the Col-R strains. The fractional inhibitory concentration index (FICI) ranged from 0.012 to 0.375, which is quite impressive. Moreover, compared with COL or kaempferol alone group and control group, colistin combined with kaempferol could efficiently inhibit biofilm formation in more than half of the Col-R strains. Further, the study was conducted to determine the in vivo therapeutic effect of the colistin/kaempferol combination against Col-R GNB strains. A G. mellonella survival assay was carried out to do this. The results showed that the combination of COL and KP was more effective than when either was used alone [208]. The antibacterial activity of three synthetic compounds—propyl gallate, propyl-, and heptyl paraben—as well as two naturally occurring phenolic compounds—kaempferol and resveratrol—against two strains of Enterococcus faecalis was examined. When combined, kaempferol and resveratrol cause a growth inhibition that is characterized by an increase in the lag phase and a decrease in the maximum specific growth rate. Moreover, kaempferol had an antagonistic effect on antioxidant activity when coupled with resveratrol, propyl- or heptyl paraben, and propyl gallate [209]. The effects of chrysin as well as kaempferol combination treatment on septic mice were examined using a 7-day survival study. Chrysin and kaempferol had different impacts on various pathophysiological elements brought on by sepsis, but they also had some synergistic effects. The kaempferol/chrysin combination had antioxidant and anti-inflammatory properties, which translated to a notable improvement in the survival rate of septic animals, even though it did not produce significant antibacterial activity. The results of this study indicate that combining these treatments could be a helpful strategy in controlling sepsis [210]. The experiment was run to assess the antibacterial properties of kaempferol and (-)-epicatechin individually and their combined effect in an in vitro setting. Both (-)-epicatechin and kaempferol had antibacterial properties, with (-)-epicatechin being more effective than kaempferol. (-)-Epicatechin and kaempferol demonstrated antibacterial qualities and protective effects against *H. pylori* infection, both individually and in combination [211]. According to a recent study, pretreatment with EGCG and kaempferol together showed protective effects by suppressing the generation of reactive oxygen species (ROS) in a dose-dependent manner and upregulating the activities of cellular antioxidant enzymes, such as glutathione peroxidase (GSH-Px), superoxide dismutase (SOD), and catalase (CAT). The mechanism underlying the synergistic antioxidant effects of kaempferol and the green tea ingredient EGCG may be the up-regulation of additional antioxidant enzyme activities, which enhance antioxidant capabilities and balance oxidative stress in cells [212].

The inhibitive effect of tarceva and kaempferol on the epidermal growth factor receptor tyrosine kinase (EGFR-TPK) and their synergistic effect on apoptosis in ovarian cancer cells SKOV-3 were examined. Kaempferol and tarceva have shown the ability to induce apoptosis of ovarian cancer cells SKOV-3 in a dose and time-dependent manner. Moreover, tarceva and kaempferol could significantly enhance caspase-3 activity in ovarian cancer cells SKOV-3 cells [213]. Another study result exhibited that combined treatments with quercetin and kaempferol were more efficient than the improved effects of each flavonol. The reduction in cell proliferation was linked with reduced expression of nuclear proliferation antigen Ki67 as well as reduced total protein levels in treated cells relative to controls [214]. According to a liver cancer study, doxorubicin and kaempferol together had a stronger inhibitory effect on the ability of liver cancer cells to proliferate. Additionally, combination therapy had superior suppressive effects on mitochondrial function, colony formation, survival, DNA damage response, and cell cycle progression. Furthermore, combination therapy also showed stronger inhibitory activity in blocking the migration as well as invasion of these cancer cells [215]. Kaempferol has been found to work in synergy with 5-fluorouracil in inhibiting cell proliferation and inducing apoptosis in colorectal cancer cells. This is accomplished through either the suppression of thymidylate synthase or the attenuation of p-Akt activation [216]. The antiproliferative potential of curcumin as well as kaempferol was examined using the DLD-1 colon cancer cell line of epithelial origin. Even the lowest concentrations of kaempferol combined with curcumin meaningfully prevented colon cancer cell proliferation. Results of this finding advocate that a combination of curcumin as well as kaempferol has important inhibitory effects on the proliferation of colon cancer cells [217].

## 7. Dietary Sources, Comparison of Efficacies with Other Treatment/Drugs, and Clinical Studies of Kaempferol

Kaempferol is present in numerous plant sources, including leaves and fruits (Figure 9). It has been recognized in numerous edible plants [219,220]. The green leafy vegetables such as spinach, cabbage, and broccoli are the richest plant sources of kaempferol. They contain 55 mg/100 g, 47 mg/100 g, and 7.2 mg/100 g of kaempferol, respectively. Other sources of kaempferol include onions with 4.5 mg/100 g and blueberries with 3.17 mg/100 g. Moreover, black tea and red wine are rich sources of kaempferol. Kaempferol concentrations are 1.7 mg/100 mL and 0.23 mg/100 mL, respectively. Also, spices like cloves, cumin, and capers contain kaempferol. Kaempferol content in capers is 104.29 mg/100 g, but cloves and cumin yield 38.6 and 23.8 mg/100 g of kaempferol, respectively [1]. The leaves of wild leeks or ramps (100 g fresh weight) were described to hold 32.5 mg of kaempferol [221]. Kaempferol quantity in other plant sources are gooseberry yellow (16 mg/kg) [222], papaya shoot (453 mg/kg), pumpkin (372 mg/kg), white radish (38 mg/kg) [219], and beans (14 mg/kg) [223]. 

There are no established dietary recommendations for flavonol intake for individuals [224,225]. A study was performed to measure the intakes of individual, classes, and total flavonoids in US adults to assess the impact of socio-demographic factors on flavonoid consumption patterns and identify major dietary sources of flavonoids. US adults consume an average of 344.83 ± 9.13 mg/day of flavonoids, with flavan-3-ols being the most prevalent class at 191.99 ± 6.84 mg/day [226]. Moreover, quercetin was the foremost flavonol consumed (13.48 0.6 mg/day), followed by kaempferol (5.38 0.22 mg/day), myricetin (3.29 0.22 mg/day) and isorhamnetin (0.49 0.02 mg/day) [226]. The in vivo and in vitro studies have confirmed that kaempferol plays a significant role in the management of pathogenesis through the modulation of inflammation, oxidative stress, and other biological activities. Clinical trials/studies on kaempferol are limited. A recent study was performed to assess the safety of administering a high dose of kaempferol aglycone with kaempferol aglycone-containing supplements to healthy adults. This study had a randomized, double-blind, placebo-controlled design and a 4-week duration. Participants were arbitrarily allocated to the kaempferol (*n* = 24) or placebo (*n* = 24) group. During the 4-week study, the kaempferol group was given a daily capsule containing 50 mg of KMP, which is roughly five times higher than the average human dietary intake. No significant differences were found in anthropometric and blood pressure measurements or blood and urine parameters between the kaempferol group and the placebo group. Additionally, there were no negative events resulting from the administration of KMP aglycone. The study results showed that healthy adults could safely consume 50 mg of kaempferol aglycone daily for four weeks [227]. Investigators have studied the efficacy of kaempferol in the management of pathogenesis in comparison with other treatments/drugs. Mohammad Hassan Jokar et al. experimented with a study that compared the molecular mechanism induced by kaempferol as well as epigallocatechin gallate (EGCG) and all-trans retinoic acid in HL-60 leukemia cells. It was reported that kaempferol reduced cell viability and increased apoptosis in these cancer cells more than EGCG. Kaempferol (concentrations of 25–100 μM) and EGCG (concentrations of 50 and 100 μM) meaningfully reduced the proliferation of HL60 cells on day 5. Likewise, a decrease in proliferation was noticed in cells treated with ATRA (10 μM). The findings of this study demonstrated that kaempferol, compared to EGCG, additionally prevented cell growth and induced apoptosis in cancer cells by increasing Bax/Bcl2 and inhibiting MDR. As compared to the three-day treatment, the IC50 of kaempferol was meaningfully reduced, while the IC50 of EGCG enhanced after five days [228]. Another study result reported that kaempferol, quercetin, as well as myricetin intensely reduced the phosphorylation of Met induced by HGF at a concentration of 20 μmol/L, whereas quercitrin did not show an effect. The half-maximal inhibitory effect (IC50) of the numerous flavanols on HGF-induced Met phosphorylation was ∼6 μmol/L for myricetin and kaempferol and 12 μmol/L for quercetin. However, quercitrin has not inhibited HGF-induced Met phosphorylation even at 20 μmol/L. Moreover, kaempferol and quercetin, at very low concentrations, effectively inhibited Akt phosphorylation induced by the activation of Met signaling. The prevention of Akt phosphorylation happens with an IC50 of 5 μmol/L for kaempferol and 2.5 μmol/L for quercetin, while quercitrin as well as myricetin did not inhibit this process. The inhibition of Akt phosphorylation by quercetin was lesser than that seen with EGCG [229]. Kaempferol, quercetin, and myricetin were examined for their role in the proliferation and survival of ovarian cells cultured in vitro. Kaempferol was found to be the most powerful bioactive compound, followed by quercetin and myricetin, at particular doses of 10, 10, and 25 μM, correspondingly [230]. The effects of the flavonols kaempferol and quercetin on the expression of intercellular adhesion molecule-1 (ICAM-1), vascular cell adhesion molecule-1 (VCAM-1), inducible NO synthase (iNOS), endothelial cell selectin (E-selectin), and cyclo-oxygenase-2 (COX-2), and on the activation of the signaling molecules NF-κB as well as activator protein-1 (AP-1), induced by a cytokine mixture in cultured human umbilical vein endothelial cells was investigated. Inhibition of reactive oxygen and nitrogen species generation did not differ between both flavonols at 1 μmol/l. However, the inhibition was significantly stronger for kaempferol at 5–50 μmol/l. The expression of adhesion molecules was always more powerfully inhibited in kaempferol-treated than in quercetin-treated cells. The inhibitory effect on COX-2 and iNOS protein levels was powerful for quercetin at 5–50 μmol/l [231]. 

## 8. Safety and Toxicity Level of Kaempferol

There are some contradictory views on the safety and toxicology of kaempferol. Some studies reported that it is safe for consumption, while others have raised apprehensions about its possible toxicity. A study was performed to examine the safety of kaempferol following subacute exposure in mice. For 28 days, kaempferol was taken orally in three separate doses. Following a 28-day dose of kaempferol, there were no treatment-related alterations in body weight or organ weights relative to the control group, nor any clinical indications of toxicity. The hematological parameters—platelet count, red blood cell, hematocrit, white blood cell, hemoglobin (Hb) level, mean corpuscular hemoglobin concentration, platelet distribution width, and red cell distribution width—showed no differences between the treated and control groups. Histopathological examination of the liver, kidney, heart, and lungs showed no morphological anomalies or lesions in either the treated or control groups [232]. An oral once-daily 13-week sub-chronic toxicity experiment involving 500, 1000, or 2000 mg/kg/day of the kaempferol aglycone-rich material was conducted on Sprague-Dawley rats. It is acceptable to use as food up to 2000 mg/kg/day because it does not cause any negative effects [233]. It is interesting to note that kaempferol can have both antioxidant and pro-oxidant effects, with the latter potentially playing a significant role in its genotoxic effects [234]. The valuable work attempted to examine the biotransformation of kaempferol and identify the CYPs that participated in its biotransformation to the more potent genotoxicant quercetin. The findings on the induction of chromosomal aberrations, as well as micronuclei in V79 cell lines genetically engineered for the expression of different rat CYP, suggest that CYP 1A1, among the cytochromes studied, is the one that plays the significant role in the biotransformation of kaempferol to the more potent genotoxicant [235]. Kaempferol was tested for carcinogenicity in rats, and kaempferol (0.04%) or a control diet was given to ACI rats for 540 days. Kaempferol was not revealed to be carcinogenic to rats [219,236]. Different studies have yielded conflicting results regarding the potential effects of kaempferol. Some researchers have reported that this flavonoid shows antimutagenic potential, whereas others have found it to be genotoxic [237,238,239,240]. A study examined the effects of cytochromes P450 in the induction of chromosomal aberrations via kaempferol in V79 cells. The findings advocate that there is a time-dependent biotransformation of kaempferol to quercetin by cytochromes P450. With the presence of microsomal metabolizing systems, quercetin seems to enhance the mutagenicity of kaempferol [241]. Some of the negative effects of kaempferol include reacting with iron, reducing folic acid uptake, reducing iron bioavailability, and increasing the bioavailability and toxicity of the anticancer drug [242,243,244]. In a 4-week randomized, double-blind clinical trial, participants were divided into a group that received 50 mg of kaempferol daily and a placebo group. No significant changes were observed in clinical measurements such as anthropometric and blood pressure or blood and urine parameters in the kaempferol group compared to the placebo group. Additionally, no adverse events owing to this flavonoid aglycone administration happened. The study results showed that consuming 50 mg of kaempferol aglycone daily for 4 weeks is safe in healthy adults [227].

## 9. Conclusions and Future Direction

Kaempferol, a flavonoid found in fruits and vegetables, has strong pharmacological properties that have been extensively researched in recent years. This flavonoid has been studied for its potential preventive role in various pathogenesis, such as diabetes, obesity, arthritis, glaucoma, wound healing, osteoporosis, and skin diseases. In addition, its role as hepatoprotective, renoprotective, cardioprotective, neuroprotective, and gastroprotective has been confirmed through the modulation of anti-oxidant, anti-inflammatory, and maintenance of tissue architectures. Furthermore, its role in cancer management has been confirmed based on in vivo and in vitro studies through the modulation of various cell signaling pathways [95]. Additionally, kaempferol, in combination with other drugs/natural compounds, synergistically inhibits disease processes by synergistically enhancing the efficacies of drugs, decreasing cell viability, and modulating the cell signaling pathways. It has been reported that kaempferol inhibits the microorganism action and effectively stops antibiotic resistance against various pathogens. This flavonoid has been broadly studied for its health-promoting effects, but it faces big obstacles, such as poor aqueous solubility, rapid metabolism, and rapid elimination from the body. Plasma exposure to this flavonoid is limited by poor oral bioavailability and, to a great extent, metabolism. This compound is quickly eliminated so that effective concentrations at the action site do not appear to be reached [245]. To overcome these problems, different approaches have been developed to improve the absorption and efficacy of this flavonoid. In this regard, innovative approaches, including nanotechnology and encapsulation methods, may enhance the delivery of this compound to target tissues, thus increasing its therapeutic efficiency [246]. Still, more studies are required to overcome the limitations of the poor aqueous solubility, rapid metabolism, and rapid elimination of kaempferol. Although preclinical research on the health benefits of kaempferol is promising, its clinical application is still limited or has a long way to go. More studies based on clinical trials are needed to utilize the health-promoting benefits of this compound and to confirm its long-term safety, efficacy, safe dose, and mode of action in the management of different pathogenesis.

## Figures and Tables

**Figure 1 molecules-29-02007-f001:**
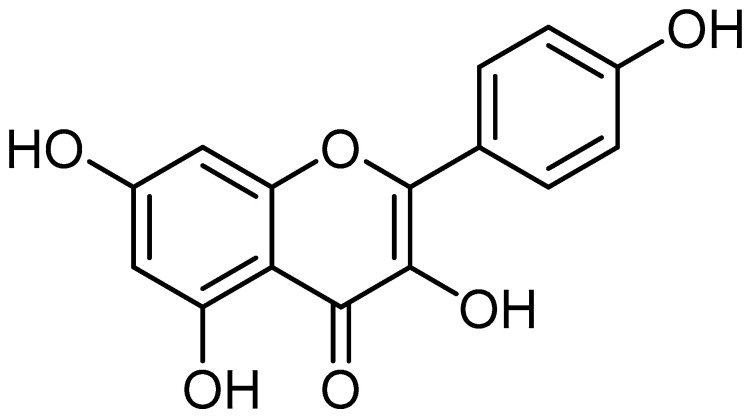
Kaempferol chemical structure (the structure was drawn using ChemDraw professional 15.0).

**Figure 2 molecules-29-02007-f002:**
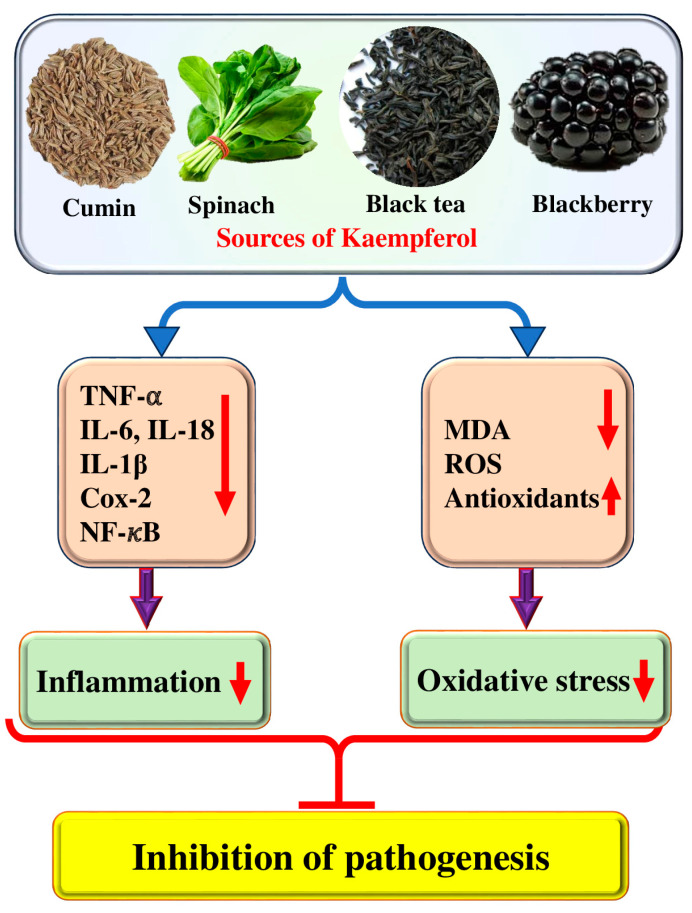
Role of kaempferol in disease management through inhibition of inflammation and oxidative stress. The upward pointing arrow indicates upregulation and the downward pointing arrow indicates downregulation.

**Figure 3 molecules-29-02007-f003:**
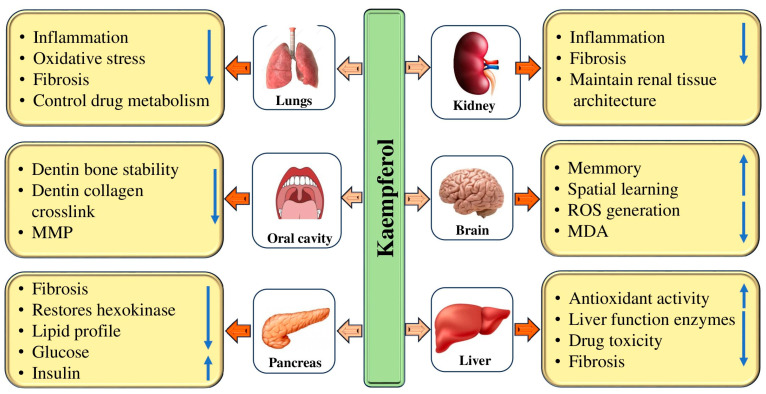
Role of kaempferol in the management of pathogenesis through modulation of biological activities. The upward pointing arrow an increase and the downward pointing arrow indicates decrease.

**Figure 4 molecules-29-02007-f004:**
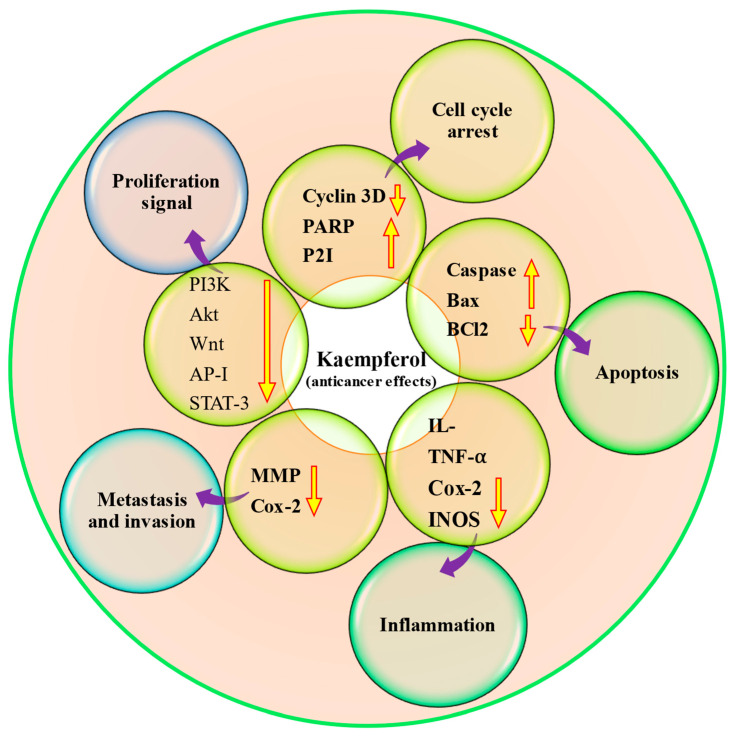
Anti-cancer potential of kaempferol through modulation of different cell signaling molecules. The upward pointing arrow indicates upregulation and the downward pointing arrow indicates downregulation.

**Figure 5 molecules-29-02007-f005:**
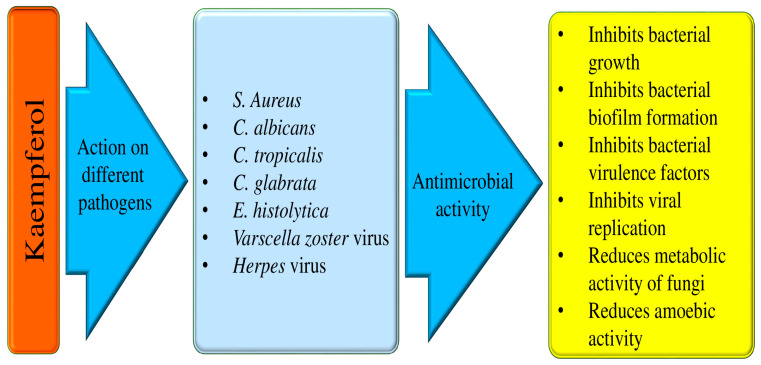
Antimicrobial properties of kaempferol.

**Figure 6 molecules-29-02007-f006:**
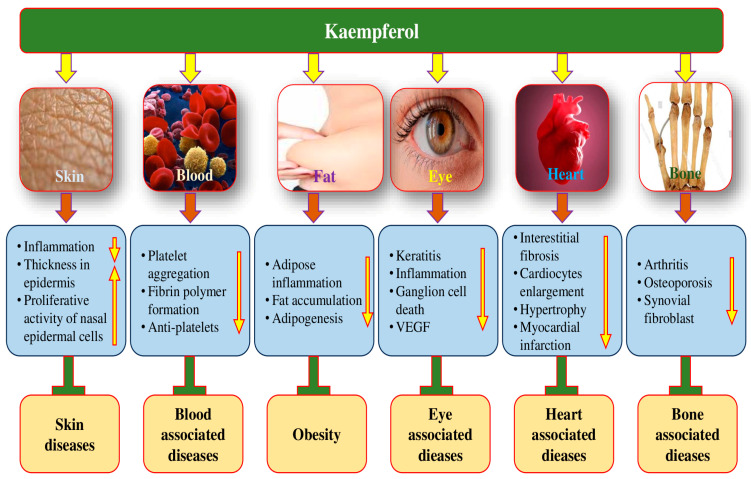
The role of kaempferol in different pathogenesis through the modulation of different biological activities. The upward pointing arrow indicates an increase and the downward pointing arrow indicates decrease.

**Figure 7 molecules-29-02007-f007:**
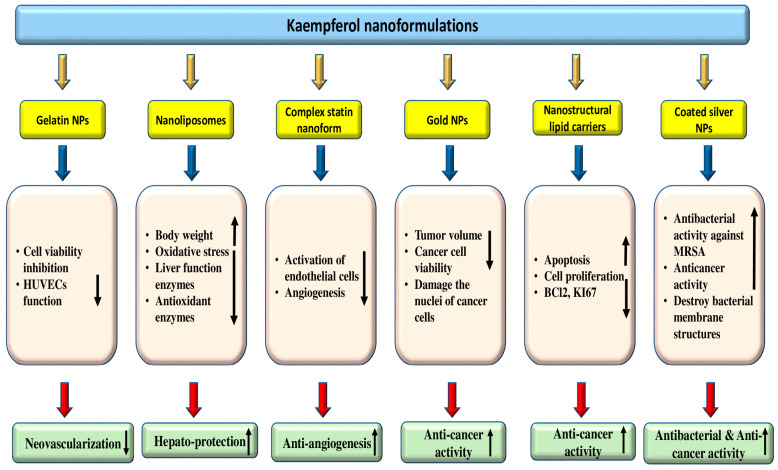
Kaempferol-based nano-formulations and their role in the management of pathogenesis. The upward pointing arrow indicates an increase and the downward pointing arrow indicates decrease.

**Figure 8 molecules-29-02007-f008:**
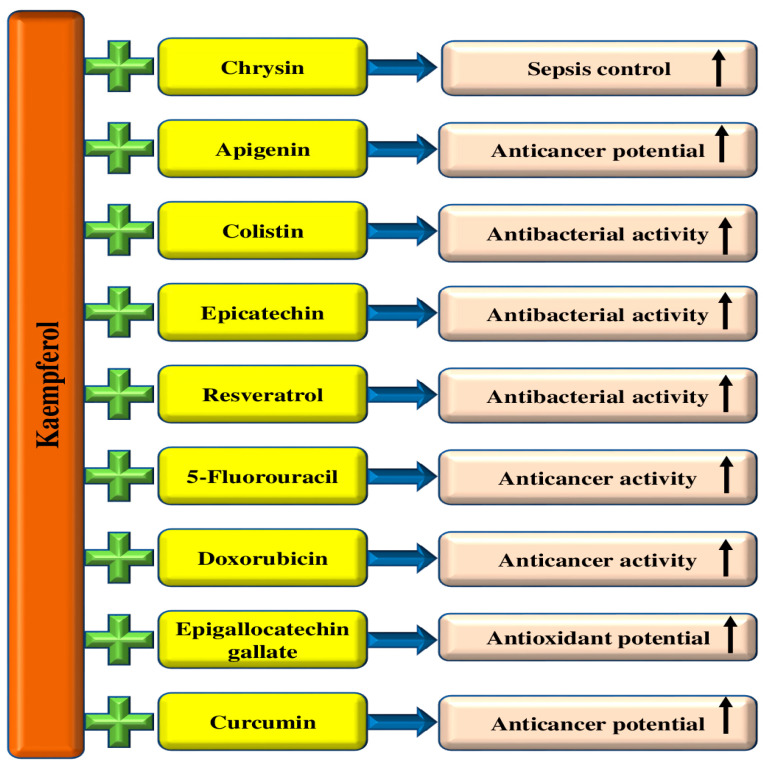
Synergistic effect of kaempferol with other natural compounds/drugs. The upward pointing arrow indicates an increase.

**Figure 9 molecules-29-02007-f009:**
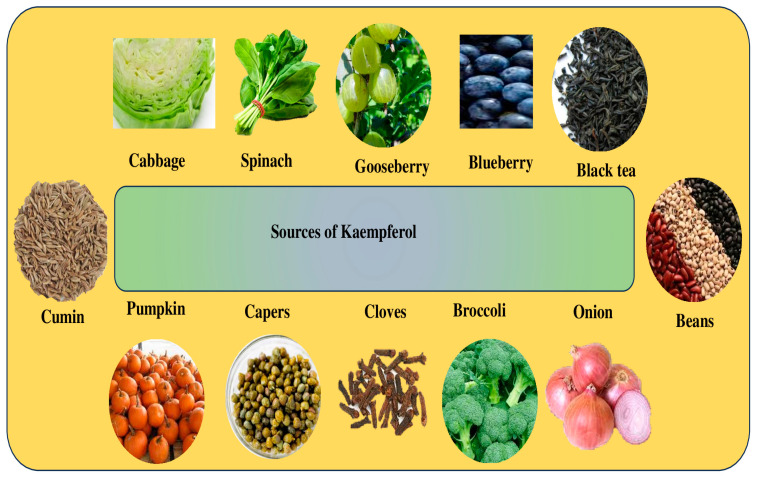
Dietary sources of kaempferol.

**Table 2 molecules-29-02007-t002:** Anti-diabetic properties of kaempferol.

Anti-diabetic potential	**Dose**	**Mechanism**	**Outcome of the Study**	**Refs.**
	50 mg/kg/day	Gluconeogenesis hepatic pyruvate carboxylase prevents	Kaempferol promotes glucose metabolism and inhibits gluconeogenesis	[43]
	50 mg/kg/day	Fasting glucose level decreased and improves insulin sensitivity	Kaempferol suppresses glucose production and improves insulin sensitivity	[44]
	50 or 150 mg/kg	Restored insulin resistance	Kaempferol has been observed to have a dose-dependent effect on ameliorating blood lipids and insulin. It has been found to effectively restore insulin resistance	[45]
	200 mg/kg	Decrease in fasting glucose and an increase in fasting insulin	Kaempferol prevents diabetic nephropathy	[46]
	100 and 200 mg/kg	Enhanced insulin and reduced blood glucose	Kaempferol elevates the release of insulin and reduces the blood glucose level	[47]
	100 mg/kg	Reversal of ATPases, Mg(2+)-ATPase, Na(+)/K(+)-ATPase	Kaempferol restore the deranged activity of membrane-bound ATPases	[48]
	100 mg/kg	Decreased lipid peroxidation and increased antioxidant	Kaempferol has been observed to decrease lipid peroxidation and increase antioxidant levels.	[28]

**Table 5 molecules-29-02007-t005:** Anti-cancer potential of kaempferol through modulation of different cell signaling molecules.

Cancer	Cell Lines	Dose	Mechanism	Study Findings	Refs.
Breast	ZR-75-30 and BT474	10, 25, 50, 100 μM	Modulation of apoptosis	Treatment with kaempferol reversed the effects of enhanced IQGAP3, which prevented cancer cells from undergoing apoptosis	[99]
Breast	MCF7	20, 40, 80 μM	Modulation of apoptosis	Kaempferol inhibition of cancer cell growth by downregulation of Bcl2 expression and inducing apoptosis	[100]
Ovarian	A2780/CP70	40 μM	Regulation of cell cycle	Kaempferol induced G2/M cell cycle arrest through the Chk2/p21/Cdc2 pathway and Chk2/Cdc25C/Cdc2 pathway	[101]
Ovarian	A2780/CP70	40 μM	Modulation of apoptosis	The late apoptotic rate of cancer cells was increased when treated with kaempferol	[101]
Breast	MDA-MB-231	40 μM	Inhibition of invasion	Kaempferol inhibits migration, adhesion, and invasion of breast carcinoma	[102]
Ovarian	OVCAR-3	20 μM	Inhibition of cMyc gene	Kaempferol treatment also inhibited cMyc gene transcription down	[103]
Ovarian	OVCAR-3	20 μM	Inhibition of angiogenesis	Kaempferol reduced VEGF secretion in OVCAR-3 cells	[104]
Bladder	EJ	40 μM	Activation of PTEN	Kaempferol increased PTEN expression	[105]

**Table 8 molecules-29-02007-t008:** Kaempferol-based nano-formulations manage the pathogenesis.

Nanoformulation	Activity	Findings	Refs.
Silver-conjugated kaempferol and hydrocortisone nanoparticles	Anti-bacterial	Bactericidal properties against different bacterial strains	[191]
Silver nanoparticle/kaempferol composites	Anti-bacterial	This formulation destroyed the membrane structure of bacteria and bacteria death	[192]
Kaempferol-chitosan/silver nanocomposite	Anti-cancer and anti-bacterial	This formulation showed inhibitory properties against triple-negative breast cancer and antibacterial property	[193]
AgNPs incorporated with kaempferol	Anti-bacterial	AgNPs-K possesses antibacterial activity against MRSA	[194]
Kaempferol-coated AgNPs	Anti-cancer	This formulation showed potential anti-cancer effects in liver cancer cells	[195]
Kaempferol-gold nanoparticle	Anti-cancer	This formulation caused a dose and time-dependent decrease in the viability of breast cancer and induced apoptosis	[196]
Kaempferol-gold nanoparticle	Anti-cancer	The formulation used in the study mainly targeted and damaged the nuclei of cancer cells. This nanocluster exhibits higher toxicity to lung cancer cells	[197]
PEGylated gold nanoparticles-DOX@Kaempferol	Anti-cancer	This formulation decreases tumor volume	[198]
Monodispersed gold nanoparticles in kaempferol	Anti-leishmanial	Nanoparticles synthesized showed a role in leishmanial chemotherapy	[199]
PEG-AuNPs@ sorafenib/kaempferol	Anti-cancer	The formulation used in the study was found to be more effective than sorafenib alone in treating cancer cells	[200]
Kaempferol-loaded nanoparticles	Anti-cancer	Pre-treatment with this formulation decreased the elevated serum levels of alanine transaminase and aspartate transaminase Additionally, the level of lipid peroxidation was attenuated	[201]
Gelatin nanoparticles with kaempferol encapsulation	Corneal neovascularization	Corneal NV treated by eye drops containing GNP-KA showed better therapeutic effects This formulation showed the capacity to inhibit the cell viability and function of HUVECs	[202]
Nano-formulated water-soluble combretastatin and kaempferol	Anti-angiogenesis	This compound has the ability to suppress angiogenesis by preventing the activation of endothelial cells and suppressing factors of angiogenesis.	[203]
Kaempferol loaded in nanostructured lipid carriers	Anti-cancer	Co-administration of KAE-loaded nanoparticles and paclitaxel strengthens the percentage of apoptosis	[204]
Nanoliposome-encapsulated kaempferol	Hepatoprotective	Dietary supplementation with this formulation improved the body weight, hepatic oxidative stress, liver enzyme activities, and antioxidant potential of the liver	[205]

**Table 9 molecules-29-02007-t009:** Synergistic effect of kaempferol with other natural compounds/drugs.

Compound	Drugs/Compound	Study Type	Activity	Outcome	Refs.
Kaempferol	Colistin	In vitro	Anti-bacterial	In vitro, the combination of colistin and kaempferol has synergistic antibacterial potential. When combined with colistin, this substance may prevent the growth of bacterial biofilms.	[208]
		In vivo	Bactericidal	When these two drugs are taken together, they have positive synergistic therapeutic effects against Col-R GNB infections.	
	Resveratrol	In vitro	Anti-bacterial	The combination of kaempferol and resveratrol inhibits growth.	[209]
	Chrysin	In vivo	sepsis control	The combination treatment of kaempferol and Chrysin boosted the overall 7-day survival rate by two times, to 29%. This combination demonstrated some promise for synergy by addressing distinct pathophysiological mechanisms associated with sepsis.	[210]
	Epicatechin	In vitro	Antibacterial	(−)-Epicatechin and kaempferol have antibacterial activities and a preventive effect against *H. pylori* infection, both when used separately and together.	[211]
	Epigallocatechin gallate	In vitro	antioxidant potential	The mechanism of synergistic antioxidant potential of EGCG in combination with kaempferol might be due to the enhancement of higher antioxidant enzyme activities.	[212]
	Tarceva	In vitro	Anti-cancer	The proliferation of ovarian cancer cells was shown to be inhibited by the combination of kaempferol and tarceva.	[213]
	Quercetin	In vitro	Anti-cancer	The combined effects of quercetin and kaempferol proved to be more effective than the individual flavonols’ additive effects. The reduction in cell proliferation was linked with reduced expression of nuclear proliferation antigen Ki67 and reduced total protein levels in treated cells relative to controls.	[214]
	Doxorubicin	In vitro	Anti-cancer	Doxorubicin and this flavonoid compound together demonstrated a more potent inhibitory effect on the viability of liver cancer cells. Combination therapy also increased the suppression of DNA damage response, colony formation, survival, cell cycle progression, and mitochondrial function.	[215]
	5-fluorouracil	In vitro	Anti-cancer	Kaempferol and 5 fluorouracil worked in concert to suppress cell proliferation and induce apoptosis.	[216]
	Curcumin	In vitro	Anti-cancer	Kaempferol and curcumin together have significant inhibitory effects on colon cancer cell proliferation.	[217]
	Apigenin	In vivo	Anti-osteoarthritis	In the rat model of ACLT-induced OA, kaempferol and apigenin may improve the effectiveness of osteoarthritis cell therapy.	[218]

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
