# Peer review of "Pharmacological Potential of Kaempferol, a Flavonoid in the Management of Pathogenesis via Modulation of Inflammation and Other Biological Activities"

_molecules, 2024, doi:10.3390/molecules29092007_

Round 1
Reviewer 1 Report
Comments and Suggestions for Authors
Dear Authors,
The paper addresses a compelling topic and is well-structured. However, I have several points and recommendations that warrant consideration:
Firstly, it would be beneficial for the authors to highlight details about the databases used for data collection/extraction (e.g., Web of Science, Scopus, Google Scholar) and the keywords employed during the literature search, alongside the period of studies included in the review. This ensures comprehensive coverage of recent and relevant studies, preferably in the introduction section.
Secondly, augmenting the paper with additional information regarding kaempferol and its promising effects against herpesviruses could enhance its scope. Herpesviruses are pathogens known to induce various diseases ranging from skin diseases to cancer and inflammatory disorders. This information can be included in the section titled "Antiviral Activity". The authors can draw from the references (DOI: 10.3390/v15122340 and DOI: 10.3390/v14030592) to extract relevant insights.
Thirdly, it's essential to address the challenges and future directions associated with utilizing kaempferol as a promising therapeutic agent in managing various diseases reviewed in this paper. This could include discussing potential limitations, such as bioavailability issues or resistance mechanisms, as well as opportunities for further research and clinical translation.
Lastly, a thorough proofreading of the full text to identify and rectify any typographical errors is recommended. Additionally, ensure that all scientific names are properly italicized for adherence to scientific writing conventions
Comments on the Quality of English LanguageThe English usage needs major refinement.
Author Response
The point-by-point response to comments and suggestions of the reviewer
Comment: The paper addresses a compelling topic and is well-structured. However, I have several points and recommendations that warrant consideration:
Response: We would like to extend our heartfelt thanks to the reviewer for thoroughly reviewing the manuscript and for providing valuable feedback and suggestions. We have taken into account each of the comments and suggestions and addressed them in a thorough and appropriate way. Consequently, we believe that the manuscript's quality and clarity have significantly improved after modification. Again, we would like to express our gratitude to the reviewer for providing valuable suggestions and comments. The changes are highlighted within the manuscript in yellow color. The point-by-point response to the reviewers’ comments and suggestions is below.
Comment: Firstly, it would be beneficial for the authors to highlight details about the databases used for data collection/extraction (e.g., Web of Science, Scopus, Google Scholar) and the keywords employed during the literature search, alongside the period of studies included in the review. This ensures comprehensive coverage of recent and relevant studies, preferably in the introduction section.
Response: We are very thankful to the reviewer for valuable comments and suggestions. As per the suggestions, databases used for data collection/extraction have been added as headings (2. Methodology) and highlighted in yellow color text. Recent and relevant studies are in the introduction section.
Comment: Secondly, augmenting the paper with additional information regarding kaempferol and its promising effects against herpesviruses could enhance its scope. Herpesviruses are pathogens known to induce various diseases ranging from skin diseases to cancer and inflammatory disorders. This information can be included in the section titled "Antiviral Activity". The authors can draw from the references (DOI: 10.3390/v15122340 and DOI: 10.3390/v14030592) to extract relevant insights.
Response: Thank you for your suggestion. As per the suggestion, kaempferol and its promising effects against herpesviruses are added as (anti-viral activities subheading 4.11-IV) and highlighted as yellow color text. Suggested references cited.
Comment. Thirdly, it's essential to address the challenges and future directions associated with utilizing kaempferol as a promising therapeutic agent in managing various diseases reviewed in this paper. This could include discussing potential limitations, such as bioavailability issues or resistance mechanisms, as well as opportunities for further research and clinical translation.
Response: Thank you for the suggestion. The conclusion section has been modified, and suggestions included properly.
Comment. Lastly, a thorough proofreading of the full text to identify and rectify any typographical errors is recommended. Additionally, ensure that all scientific names are properly italicized for adherence to scientific writing conventions.
Response: Thank you for your valuable suggestion. We have carefully checked all errors and corrected them accordingly.
Reviewer 2 Report
Comments and Suggestions for Authors
The authors describes the pharmacological potential of kaempferol.
Some suggestions to improve this manuscript.
Title: I suggest: Pharmacological Potential of Kaempferol in the Management of Pathogenesis and other Biological Activities
Abstract: Correct in vivo and in vitro to italic.
Introduction: I suggest remove the lines 27-32. Begin the text with Kaempferol.
Bioavailability: The pharmacokinetics of flavonoids have been widely studied in vitro and in vivo. Put in vivo and in vitro to italic. In other parts also
Figure 3 needs improvement of quality
I suggest put antimicrobial effects at the end of the revision. Is confusing antimicrobial effect and anti-artritis effects in sequence
Figure 7 needs to be improved.
The pharmacokinetics of flavonoids have been widely studied in vitro and in vivo
Author Response
The point-by-point response to comments and suggestions of the reviewer
Comment: The authors describe the pharmacological potential of kaempferol.
Some suggestions to improve this manuscript.
Title: I suggest: Pharmacological Potential of Kaempferol in the Management of Pathogenesis and Other Biological Activities
Response: We are very thankful to the reviewer for the thorough review, valuable comments, and suggestions. We tried our best to respond to the comments properly and fulfilling all these valuable comments and suggestions has definitely improved the quality of this manuscript.
We kept the title as per the special issue of the journal, which is focused on inflammation, hence we did not change the title
Comment: Abstract: Correct in vivo and in vitro to italic.
Response: Thank you for your suggestion. As per your suggestion, the in vivo and in vitro sections of the manuscript have been corrected and highlighted in yellow colr.
Comment: Introduction: I suggest remove the lines 27-32. Begin the text with Kaempferol.
Response: Thank you for your valuable suggestion. Lines 27-32 removed and the paragraph begins with kaempferol.
Comment: Bioavailability: The pharmacokinetics of flavonoids have been widely studied in vitro and in vivo. Put in vivo and in vitro to italic. In other parts also
Response: Thank you for your suggestion. As per suggestion, in vivo and in vitro corrected throughout the manuscript and highlighted as yellow color text.
Comment: Figure 3 needs improvement of quality
Response: Thank you for your valuable suggestion. The quality of images appropriately improved.
Comment: I suggest put antimicrobial effects at the end of the revision. Is confusing antimicrobial effect and anti-artritis effects in sequence
Response: Thank you for your suggestions. Subheading 4.11 has been added as "Anti-microbial effects," which describes the substance's anti-microbial potential. Additionally, subheading 4.15 has been added as "Anti-arthritis effects," which explains the substance's potential for treating arthritis. Subheading 4.11 is okay, and there should be no further confusion with the subheadings.
Comment: Figure 7 needs to be improved.
Response: Thank you for your valuable suggestion. The quality of images improved properly.
Comment: The pharmacokinetics of flavonoids have been widely studied in vitro and in vivo
Response: Thank you for your valuable suggestion. Corrected throughout the manuscript.
Reviewer 3 Report
Comments and Suggestions for Authors
The article covers a wide range of Kaempferol’s pharmacological activities, including its antioxidant, anti-inflammatory, anti-diabetic, and other health-promoting effects. To improve, consider expanding on each activity by providing more examples of studies or clinical trials that support these findings. Additionally, discussing any conflicting evidence or limitations of current research could provide a more balanced perspective.
Comments hereafter:
1. Ensure all claims are backed by up-to-date and high-quality sources. This includes peer-reviewed articles, clinical trials, and authoritative reviews. Each source should be cited appropriately within the text and in a reference list at the end. Review the current references for accuracy and completeness.
2. Where possible, elaborate on the mechanisms of action for Kaempferol’s various effects. This will help readers understand not just what Kaempferol does, but how it achieves its effects at a molecular or cellular level.
3. The article mentions the need for further investigations into Kaempferol’s toxicity and safety aspects. Expanding on known safety profiles, potential side effects, and any known interactions with medications or conditions could greatly improve the article's utility to readers interested in its therapeutic potential.
4. Figures and Tables: ok
5. Where research permits, include sections on the practical applications of Kaempferol, such as its inclusion in dietary sources, potential for inclusion in supplements, or ongoing research into pharmaceutical uses. Information on dosages, administration methods, and efficacy comparisons with other treatments could also be valuable.
6. Conclude with a section on future directions for Kaempferol research. Highlighting areas that require further investigation, such as its clinical applications, long-term safety, and efficacy in human populations, can underscore the importance of ongoing and future studies.
7. Ensure the article is accessible to a broad audience. This includes avoiding overly technical language where possible and defining any specialized terms that are used. A glossary or a section explaining key terms could be beneficial.
8. Each section should begin with a brief introduction outlining what will be discussed, followed by detailed paragraphs that flesh out the key points.
Comments on the Quality of English LanguageMinor editing of English language required
Author Response
The point-by-point response to comments and suggestions of the reviewer
Comment: The article covers a wide range of Kaempferol’s pharmacological activities, including its antioxidant, anti-inflammatory, anti-diabetic, and other health-promoting effects. To improve, consider expanding on each activity by providing more examples of studies or clinical trials that support these findings. Additionally, discussing any conflicting evidence or limitations of current research could provide a more balanced perspective.
Response: We would like to extend our heartfelt thanks to the reviewer for a thorough review of the manuscript and for providing valuable feedback and suggestions. We have taken into account each of the comments and suggestions and addressed them thoroughly and appropriately. Consequently, we believe that the manuscript's quality and clarity have significantly improved after modification. We would once again like to express our gratitude to the reviewer for giving valuable suggestions and comments.
Thank you for your valuable suggestion. As per suggestion, activities or findings were expanded properly and highlighted in a different section as yellow color text.
Comment: Ensure all claims are backed by up-to-date and high-quality sources. This includes peer-reviewed articles, clinical trials, and authoritative reviews. Each source should be cited appropriately within the text and in a reference list at the end. Review the current references for accuracy and completeness.
Response: Thank you for your valuable suggestion. As per the suggestion, references have been checked and cited properly. Moreover, references are updated properly.
Comment: Where possible, elaborate on the mechanisms of action for Kaempferol’s various effects. This will help readers understand not just what Kaempferol does, but how it achieves its effects at a molecular or cellular level.
Response: Thank you for your valuable suggestion. As per suggestion, mechanisms of action expanded properly and highlighted in a different section as yellow color text.
Comment: The article mentions the need for further investigations into Kaempferol’s toxicity and safety aspects. Expanding on known safety profiles, potential side effects, and any known interactions with
Response: Thank you for your valuable suggestion. As per suggestion, Kaempferol’s toxicity and safety aspects as section 8: Safety and toxicity level have been properly modified and highlighted as yellow color text.
Comment: Figures and Tables: ok
Response: Thank you
Comment: Where research permits, include sections on the practical applications of Kaempferol, such as its inclusion in dietary sources, potential for inclusion in supplements, or ongoing research into pharmaceutical uses. Information on dosages, administration methods, and efficacy comparisons with other treatments could also be valuable.
Response: Thank you for your valuable suggestion. As per suggestion, Dietary Sources, comparison of efficacies with other treatment/drugs, and clinical studies of kaempferol have been added as section 7. Dietary Sources, comparison of efficacies with other treatment/drugs, and clinical studies of kaempferol and highlighted as yellow colour text
Comment: Conclude with a section on future directions for Kaempferol research. Highlighting areas that require further investigation, such as its clinical applications, long-term safety, and efficacy in human populations, can underscore the importance of ongoing and future studies.
Response: Thank you for your valuable suggestion. As per suggestion, the conclusion section has been modified accordingly.
Comment: Ensure the article is accessible to a broad audience. This includes avoiding overly technical language where possible and defining any specialized terms that are used. A glossary or a section explaining key terms could be beneficial.
Response: Thank you for your valuable suggestion. Language has been checked properly and improved accordingly. Moreover, a section explaining key terms added in the end of the manuscript.
Comment: Each section should begin with a brief introduction outlining what will be discussed, followed by detailed paragraphs that flesh out the key points.
Response: Thank you for your valuable suggestion. A brief introduction in most of the section added.
Comment: Minor editing of English language required
Response: Thank you for your valuable suggestion. The language was checked properly and improved accordingly.
Round 2
Reviewer 1 Report
Comments and Suggestions for Authors
The msnuscript has sufficiently been improved.
Comments on the Quality of English LanguageThe English usage is fine; however, minor refinement is needed during the proofreading.